# Acetic acid activates distinct taste pathways in *Drosophila* to elicit opposing, state-dependent feeding responses

Anita V Devineni[1], Bei Sun[1], Anna Zhukovskaya[1], Richard Axel[1,2]*

[1]Department of Neuroscience, The Mortimer B Zuckerman Mind Brain Behavior Institute, Columbia University, New York, United States; [2]Howard Hughes Medical Institute, Columbia University, New York, United States

**Abstract** Taste circuits are genetically determined to elicit an innate appetitive or aversive response, ensuring that animals consume nutritious foods and avoid the ingestion of toxins. We have examined the response of *Drosophila melanogaster* to acetic acid, a tastant that can be a metabolic resource but can also be toxic to the fly. Our data reveal that flies accommodate these conflicting attributes of acetic acid by virtue of a hunger-dependent switch in their behavioral response to this stimulus. Fed flies show taste aversion to acetic acid, whereas starved flies show a robust appetitive response. These opposing responses are mediated by two different classes of taste neurons, the sugar- and bitter-sensing neurons. Hunger shifts the behavioral response from aversion to attraction by enhancing the appetitive sugar pathway as well as suppressing the aversive bitter pathway. Thus a single tastant can drive opposing behaviors by activating distinct taste pathways modulated by internal state.

DOI: https://doi.org/10.7554/eLife.47677.001

## Introduction

Gustatory systems have evolved to identify appetitive substances of nutritional value and to elicit avoidance of toxic compounds. Organisms may encounter food sources that also contain harmful substances, and this poses an interesting perceptual problem. *Drosophila melanogaster*, for example, dines on fermenting fruit that contains both appetitive and aversive compounds. During fermentation, yeast and bacteria enzymatically convert six-carbon sugars into ethanol and acetic acid. These fermentation products can be toxic to the fly, yet the scent of decaying fruit is attractive and moreover the fly feeds despite the presence of these toxic compounds (*McKenzie and Parsons, 1972*; *McKenzie and McKechnie, 1979*; *Zhu et al., 2003*). These observations suggest that adaptive mechanisms may have evolved to ensure that toxic products do not impair the hungry fly from approaching and feeding on decaying fruits. Cider vinegar, for example, contains the potentially toxic metabolite acetic acid but elicits strong odor-evoked attraction (*Semmelhack and Wang, 2009*). *Drosophila melanogaster*, termed the vinegar fly, is more resistant to the toxic effects of acetic acid than other *Drosophila* species that do not depend upon fermenting food sources and shows greater tolerance to acetic acid than structurally similar carboxylic acids (*McKenzie and McKechnie, 1979*; *Parsons, 1980*; *Chakir et al., 1993*). Moreover, flies can utilize acetic acid as a caloric source when deprived of other food sources (*Parsons, 1980*; *Hoffmann and Parsons, 1984*); acetic acid is converted into acetyl-CoA and metabolized by the tricarboxylic acid cycle. Thus *D. melanogaster* may have evolved specific adaptations that allow the fly to recognize acetic acid as an appetitive tastant despite its potential toxicity. We have explored the seemingly paradoxical effects of acetic acid on feeding behavior in the vinegar fly.

*For correspondence:
ra27@columbia.edu

**Competing interests:** The authors declare that no competing interests exist.

**eLife digest** Our sense of taste is critical to our survival. Taste helps us to consume nutritious foods and avoid toxins. There are five basic taste categories: sweet, salty, bitter, sour, and umami or savory, a taste typical of protein-rich foods. Each taste category activates a distinct pathway in the brain, triggering specific feelings and behaviors. We normally find sugar, salt, and components of protein pleasant, and seek out foods with these tastes. By contrast, we often find overly bitter or sour tastes unpleasant and try to avoid them. As sour and bitter-tasting substances often contain toxins, this response helps to protect us from poisoning.

Across the animal kingdom, these preferences are largely hardwired from birth. But the relationship between taste and nutrients is not always straightforward. Some substances can be toxic despite also containing useful nutrients. Overripe fruit, for example, is broken down by yeast and bacteria to produce acetic acid, or vinegar. Like other acids, acetic acid can be toxic. But for the fruit fly *Drosophila melanogaster*, also known as the vinegar fly, acetic acid from rotten fruit can be a valuable source of calories. So how do flies react to the taste of acetic acid?

Devineni et al. show that, unlike other chemicals, acetic acid triggers different taste responses in flies depending on whether the insects are hungry. Well-fed flies find the taste repulsive, probably because it signals toxicity. But hungry flies find it attractive, presumably because of their overriding need for calories. Devineni et al. show that acetic acid activates both sugar-sensing and bitter-sensing pathways in the fly brain. Hunger increases activity in the sugar pathway and reduces it in the bitter pathway. As a result, hungry flies are attracted to acetic acid, whereas fully fed flies are repulsed.

Flexibility in the taste system enables animals to react to the same substance in different ways depending on their current needs. Related to this, evidence suggests that obesity may be associated with altered sensitivity to certain tastes, such as sweet, as well as a blunted response to satiety signals. Understanding how the brain combines information about taste and hunger to control food consumption may ultimately help us to understand and treat obesity.

DOI: https://doi.org/10.7554/eLife.47677.002

Feeding is initiated by extension of the proboscis, a behavior that allows the fly to taste a potential food source (*Dethier, 1976*). Flies recognize a relatively small number of basic taste categories, including sweet, salty, bitter, sour, fat, and carbonation (*Liman et al., 2014*; *Zhang et al., 2013*; *Jaeger et al., 2018*; *Chen and Amrein, 2017*; *Masek and Keene, 2013*; *Fischler et al., 2007*). As in mammals, most tastants excite only one class of sensory neurons and each class is thought to activate determined neural pathways to elicit innate behavioral responses (*Liman et al., 2014*). For example, activation of sugar-responsive neurons drives appetitive feeding responses, whereas bitter-responsive neurons elicit aversion and suppress feeding (*Liman et al., 2014*; *Marella et al., 2006*).

Sour taste, evoked by acids, is less well-understood than other taste modalities. Acids are potentially toxic to animals and may also indicate that food is unripe or spoiled. Both flies and mammals generally exhibit taste aversion to strongly acidic food (*Charlu et al., 2013*; *DeSimone et al., 2001*). However, flies do not uniformly avoid acidic stimuli: they show greater sugar consumption at an acidic pH than at a neutral or basic pH (*Deshpande et al., 2015*), and at low concentration acids may counteract the repulsive effect of bitter compounds on feeding (*Chen and Amrein, 2014*). Flies also prefer to lay eggs on carboxylic acids such as acetic and citric acid (*Joseph et al., 2009*; *Chen and Amrein, 2017*). This ovipositional preference is mediated by taste sensory neurons in the legs that respond specifically to acids (*Chen and Amrein, 2017*). Dedicated acid-sensing neurons in the proboscis have not been identified, although acid responses in bitter-sensing neurons have been observed (*Charlu et al., 2013*; *Rimal et al., 2019*).

The activation of gustatory neurons in flies and mammals elicits innate behavioral responses, but these responses can be modulated by internal states such as satiety or hunger. Hunger elicits several adaptive changes in behavior: increased food-seeking and food consumption, enhanced locomotor activity, decreased sleep, and altered olfactory and taste sensitivity (*Sternson et al., 2013*; *Itskov and Ribeiro, 2013*; *Pool and Scott, 2014*; *Yang et al., 2015*). In flies, starvation increases sugar sensitivity, which promotes feeding, and decreases bitter sensitivity, which enhances

acceptance of food sources that contain aversive tastants (*Inagaki et al., 2012*; *Inagaki et al., 2014*). Starved flies also show enhanced olfactory attraction to cider vinegar, which facilitates food search behavior (*Root et al., 2011*). These hunger-dependent changes in both olfactory and gustatory sensitivity result, at least in part, from alterations in sensory neuron activity (*Inagaki et al., 2012*; *Inagaki et al., 2014*; *Root et al., 2011*).

Acetic acid, a product of fruit fermentation, signals the presence of food preferred by the fly and may also serve as a caloric source (*Parsons, 1980*; *Hoffmann and Parsons, 1984*). Acetic acid, however, can be toxic and flies avoid residing on food containing acetic acid (*Parsons, 1980*; *Joseph et al., 2009*). We have examined the behavioral and neural responses to the taste of acetic acid and observe that hunger induces a dramatic switch in the behavioral response to this metabolite. Fed flies show taste aversion to acetic acid whereas starved flies exhibit a strong appetitive response. Genetic silencing demonstrates that the bitter-sensing neurons mediate acetic acid aversion whereas the sugar-sensing neurons mediate the appetitive response to acetic acid. Hunger shifts the response from aversion to attraction by enhancing the sugar pathway as well as suppressing the bitter pathway. Calcium imaging reveals that acetic acid activates both sugar- and bitter-sensing neurons in both the fed and starved state. Thus, a single tastant activates two distinct neural pathways that elicit opposing behaviors dependent upon internal state. This hunger-dependent switch may reflect an adaptive response to acetic acid, a potential toxin that can also afford nutritional value under extreme conditions.

## Results

### Acetic acid can elicit an appetitive or aversive taste response

The taste response to acetic acid was analyzed by examining the proboscis extension response (PER), an appetitive response that initiates feeding (*Dethier, 1976*). Appetitive tastants elicit PER when applied to the legs or labellum, the distal segment of the proboscis. Aversive tastants do not elicit PER and diminish the PER elicited by an attractive tastant (*Dethier, 1976*). In fed flies, exposure of acetic acid (1–10%) to the labellum elicited very weak levels of PER similar to those induced by water (*Figure 1A*), even though these flies showed strong PER to sucrose (*Figure 1B*). Moreover, when acetic acid was mixed with either 300 mM or 50 mM sucrose it strongly reduced sucrose-evoked PER in fed flies (*Figure 1C–D*). 79% of fed flies exhibited PER to 300 mM sucrose alone, and this response was reduced to 44% when 10% acetic acid was added (*Figure 1C*). These data demonstrate that acetic acid elicits taste aversion in fed flies.

A dramatic switch was observed in the behavioral response of starved flies. When acetic acid was applied to the labellum of one- or two-day starved flies, strong, dose-dependent PER was observed, with 86% of flies exhibiting PER to 10% acetic acid after two days of starvation (*Figure 1A*). Moreover, when acetic acid was added to concentrations of sucrose ranging from 5 mM to 300 mM, no suppression of sugar-evoked PER was observed in two-day starved flies; in fact, the addition of acetic acid enhanced PER to 5 or 10 mM sucrose (*Figure 1C–D* and *Figure 1—figure supplement 1*). In one-day starved flies, acetic acid weakly suppressed PER at 50 mM but not 300 mM sucrose (*Figure 1C–D*). Thus the behavior of starved flies, especially after two days of food deprivation, contrasts sharply with the strong PER suppression induced by acetic acid in fed flies, indicating that acetic acid is aversive to fed flies but becomes appetitive after starvation.

This switch from an aversive response in fed flies to an appetitive response in starved flies is specific for acetic acid and was not observed for other aversive tastants, such as the bitter compounds quinine and lobeline (*Figure 1—figure supplement 2*). Appetitive compounds such as sugar also do not elicit a qualitative switch in behavior, since sugar elicited PER in both fed and starved flies (*Figure 1B*). Thus, acetic acid appears unique in its ability to elicit opposing behavioral responses dependent upon internal state.

We next performed experiments to demonstrate that PER elicited by acetic acid is a component of an appetitive feeding response. When a fly is stimulated asymmetrically with an appetitive tastant on only one leg, extension of the proboscis is observed in the direction of the stimulus (*Saraswati, 1998*; *Schwarz et al., 2017*). We first confirmed that acetic acid elicits strong PER in starved flies when applied to the legs instead of the labellum (*Figure 1E*). We then stimulated the legs asymmetrically and observed that in 72% of trials, starved flies that showed PER extended the proboscis

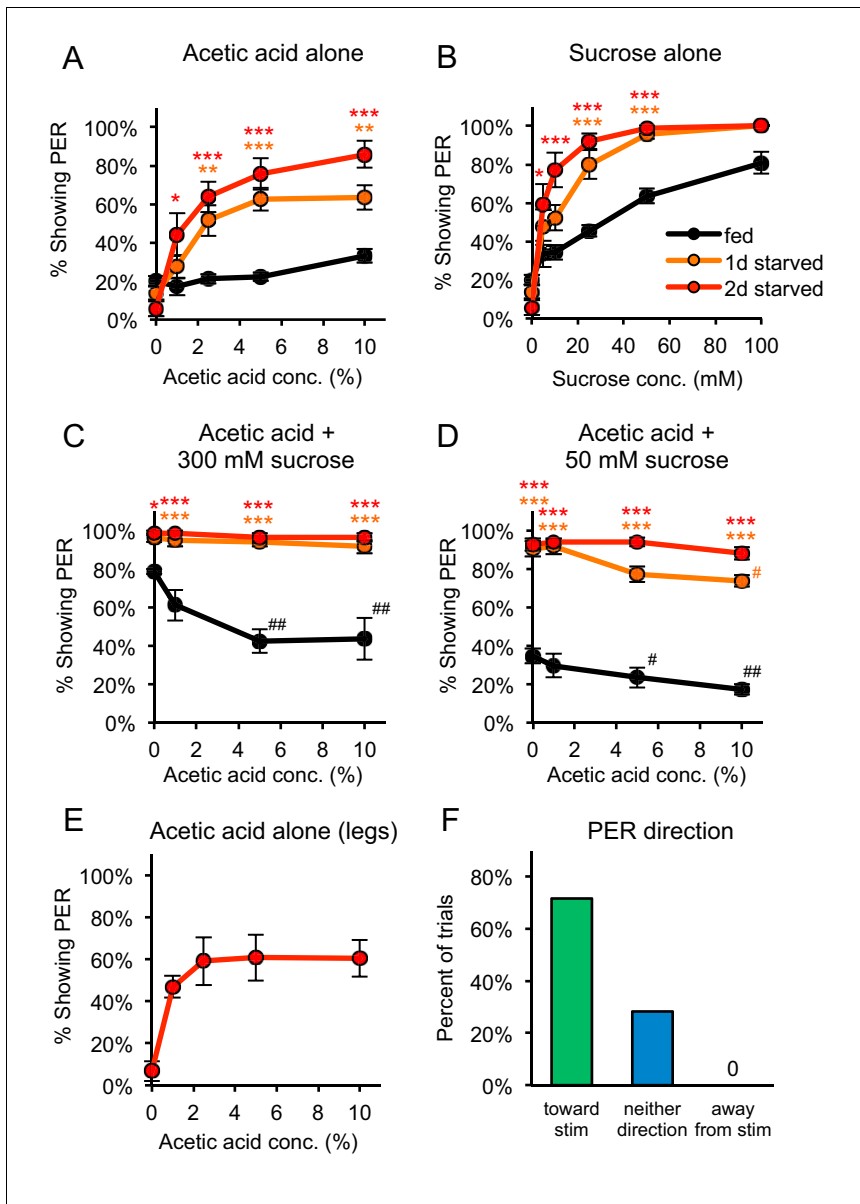

**Figure 1.** Acetic acid induces aversive or appetitive taste responses depending on hunger state. (A) One-day or two-day starved flies, but not fed flies, showed strong PER to acetic acid applied to the labellum. PER at 0% acetic acid represents the baseline response to water. (B) Both fed and starved flies showed dose-dependent PER to sucrose applied to the labellum. (C) Acetic acid suppressed PER to 300 mM sucrose in fed flies but not in one-day or two-day starved flies. (D) Acetic acid suppressed PER to 50 mM sucrose in fed flies and had a small but significant effect in one-day starved flies, but not in two-day starved flies. (E) Two-day starved flies showed PER to acetic acid applied to the legs. (F) Two-day starved flies stimulated asymmetrically with 5% acetic acid on the legs tended to show PER toward the stimulus (n = 53 trials, nine flies). In panels A-D, fed and starved flies were compared using two-way repeated measures ANOVA followed by Bonferroni's post-tests (*p<0.05, **p<0.01, ***p<0.001; orange or red asterisks correspond to one- or two-day starved flies, respectively). In panels C-D, responses within each group were compared to the response to 0% acetic acid using one-way repeated-measures ANOVA followed by Dunnett's post-tests (#p<0.05, ##p<0.01; symbols colored by group). Detailed statistical results for all experiments are reported in *Supplementary file 1*. For panels A-E, n = 3–5 sets of flies. See also *Figure 1—figure supplements 1–3* and *Videos 1* and *2*.

DOI: https://doi.org/10.7554/eLife.47677.003

The following source data and figure supplements are available for figure 1:

**Source data 1.** Raw data for *Figure 1*.

*Figure 1 continued on next page*

*Figure 1 continued*

DOI: https://doi.org/10.7554/eLife.47677.007

**Figure supplement 1.** At low sucrose concentrations acetic acid enhances sucrose-evoked PER in two-day starved flies.

DOI: https://doi.org/10.7554/eLife.47677.004

**Figure supplement 2.** Starved flies show aversion to bitter compounds.

DOI: https://doi.org/10.7554/eLife.47677.005

**Figure supplement 3.** PER to other acids and acetate in starved flies.

DOI: https://doi.org/10.7554/eLife.47677.006

in the direction of the stimulus (*Figure 1F*, *Video 1*). Proboscis extension was never observed in the direction opposing the stimulus, and in 28% of trials flies exhibited PER neither toward nor away from the stimulus (*Figure 1F*). When afforded the option to consume 5% acetic acid following PER, 7 of the 10 flies tested consumed it (*Video 2*). Thus, the majority of flies extend their proboscis in the direction of an acetic acid stimulus and voluntarily consume it, suggesting that this response is an appetitive component of feeding behavior.

Acetic acid exists in solution as three chemical species: undissociated acetic acid, which partially dissociates to produce acetate and protons. We asked whether PER to acetic acid reflects a more general taste response to low pH. Starved flies failed to show PER to hydrochloric acid at pH values equivalent to those of 5% or 10% acetic acid, which elicit strong PER, indicating that low pH is not sufficient to induce an appetitive response (*Figure 1—figure supplement 3A*). We also tested the response of starved flies to potassium acetate at molarities equivalent to those of 5% or 10% acetic acid and failed to observe a response (*Figure 1—figure supplement 3B*). These experiments suggest that neither protons nor acetate ions are capable of eliciting PER, suggesting that undissociated acetic acid is recognized by taste cells. In accord with this suggestion, propionic acid, a simple carboxylic acid structurally similar to acetic acid, elicited strong PER in starved flies whereas the more distantly related citric acid elicited a weaker response (*Figure 1—figure supplement 3C*). Thus, undissociated small aliphatic acids may be recognized by gustatory neurons to elicit PER in starved flies.

## PER to acetic acid is mediated by the gustatory system

Proboscis extension can be elicited by the taste organs, but it remains possible that other sensory modalities such as olfaction contribute to this behavioral response. Acetic acid activates olfactory sensory neurons (*Ai et al., 2010*). We therefore removed the olfactory organs, the third antennal segment and maxillary palp, from two-day starved flies and observed that PER to acetic acid was unperturbed by this manipulation (*Figure 2A*). 76% of flies lacking olfactory organs showed PER to 10% acetic acid, a value close to that observed with control flies (72%; *Figure 2A*). Fed flies lacking

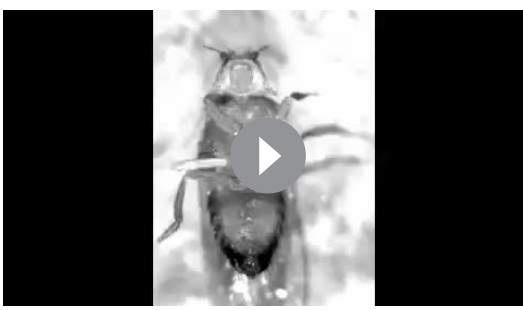

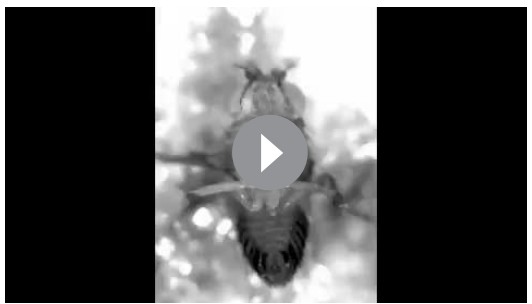

**Video 1.** Directional PER to acetic acid. A Kimwipe containing 5% acetic acid was alternately applied to the left or right legs of a two-day starved fly. Four trials (two left and two right) are shown in this video. In each case the fly extended its proboscis toward the stimulus.
DOI: https://doi.org/10.7554/eLife.47677.008

**Video 2.** Starved fly voluntarily consuming acetic acid. A Kimwipe containing 5% acetic acid was applied to the legs of a two-day starved fly, which caused the fly to exhibit PER and ingest acetic acid from the Kimwipe for approximately 7 s.
DOI: https://doi.org/10.7554/eLife.47677.009

olfactory organs failed to show PER to acetic acid, mirroring the behavior of control flies (*Figure 2B*). Fed flies lacking olfactory organs also continued to show aversion to acetic acid, as we observed significant suppression of PER when acetic acid was added to sucrose (*Figure 2C*). These experiments demonstrate that both the appetitive and aversive proboscis extension responses to acetic acid are observed in the absence of olfactory organs, and are likely to be mediated by the gustatory system.

We demonstrated the requirement for taste neurons by examining the response to acetic acid in *pox-neuro* (*poxn*) mutants, in which taste bristles are transformed into mechanosensory bristles lacking gustatory receptors (*Boll and Noll, 2002*). Starved *poxn*$^{\Delta M22-B5}$ homozygous mutants failed to display PER to any concentration of acetic acid tested (*Figure 2D*). In contrast, wild-type flies, *poxn*$^{\Delta M22-B5}$/+heterozygotes (which have normal bristles), and rescue flies (*poxn*$^{\Delta M22-B5}$ mutants carrying the *SuperA* rescue transgene; *Boll and Noll, 2002*) showed strong PER, with up to ~50–70% of flies responding (*Figure 2D*). Interpretation of these experiments must be tempered by the observation that the *poxn*$^{\Delta M22-B5}$ mutants often appeared physically smaller than control flies and are known to have central nervous system abnormalities in addition to their lack of taste bristles (*Boll and Noll, 2002*). Nonetheless, these experiments suggest that the appetitive and aversive responses to acetic acid require the taste organs and are largely independent of olfaction.

## Sugar-sensing neurons mediate PER to acetic acid

Neurons in the chemosensory bristles of the labellum detect distinct taste modalities, including sugar, bitter, water, and low and high concentrations of salt (*Liman et al., 2014*; *Marella et al., 2006*; *Cameron et al., 2010*; *Zhang et al., 2013*; *Jaeger et al., 2018*). We employed genetic silencing to identify the neuronal classes responsible for the appetitive and aversive responses to acetic acid. Sugar-sensing neurons express multiple chemoreceptors and elicit PER in response to sugars (*Dahanukar et al., 2007*; *Slone et al., 2007*; *Fujii et al., 2015*). The receptor Gr64f is expressed in all sugar-responsive taste neurons (*Fujii et al., 2015*). We therefore silenced the sugar-sensing neurons by expressing *UAS-Kir2.1*, encoding an inwardly rectifying potassium channel (*Baines et al., 2001*), under the control of the transcriptional activator *Gr64f-Gal4*. Starved flies harboring both the *Gr64f-Gal4* and *UAS-Kir2.1* transgenes showed very low frequencies of PER (12–28%) in response to increasing concentrations of either sucrose or acetic acid (*Figure 3A–B*). Control flies containing either the *Gr64f-Gal4* or *UAS-Kir2.1* transgenes alone resembled wild-type flies and exhibited strong PER to both sucrose and acetic acid, with 100% of flies responding to sucrose and ~70–80% responding to acetic acid (*Figure 3A–B*). These experiments demonstrate that the appetitive response to acetic acid observed in starved flies is mediated by the sugar-sensing neurons.

We asked whether the acetic acid response is mediated by the gustatory receptors (Grs) that detect sugars. Eight Grs are expressed in sugar sensory neurons of the labellum (*Fujii et al., 2015*; *Yang et al., 2015*). As expected, homozygous flies carrying deletions in all eight sugar-sensing Gr genes (Δ8Grs/Δ8Grs; *Yavuz et al., 2014*) showed a strong reduction in PER to sucrose as compared with control heterozygous flies (*Figure 3C*). Homozygous mutant flies did exhibit some residual PER to sucrose, which may reflect the presence of additional uncharacterized sugar receptors. In contrast to their strongly reduced response to sugar, homozygous mutant flies continued to show PER to acetic acid (*Figure 3D*). Interestingly, acetic acid-evoked PER in homozygous mutant flies was significantly greater than in control flies (*Figure 3D*). This increase in PER may reflect the possibility that the sugar-sensing circuit is upregulated in the mutants due to diminished sensory activity or intensified hunger. Alternatively, the absence of sugar receptors at the dendritic membrane may allow for increased accumulation of acetic acid receptors, or acetic acid and sucrose transduction pathways may employ a common limiting component that is no longer limiting in mutant sugar-sensing neurons. Overall, these experiments demonstrate that the response to acetic acid in starved flies is mediated by the sugar-sensing neurons but does not employ the sugar receptors.

The sugar-sensing neurons also elicit PER in response to fatty acids, such as hexanoic and octanoic acid, through a molecular mechanism distinct from sugar detection (*Masek and Keene, 2013*; *Tauber et al., 2017*; *Ahn et al., 2017*). We therefore asked whether sugar neurons recognize acetic acid and fatty acids through the same mechanism, since hexanoic and octanoic acids also are aliphatic carboxylic acids. A previous study showed that PER induced by fatty acids requires phospholipase C (PLC) signaling in sugar-sensing neurons whereas PLC is dispensable for PER to sucrose (*Masek and Keene, 2013*). We therefore tested whether PLC signaling in sugar neurons is required

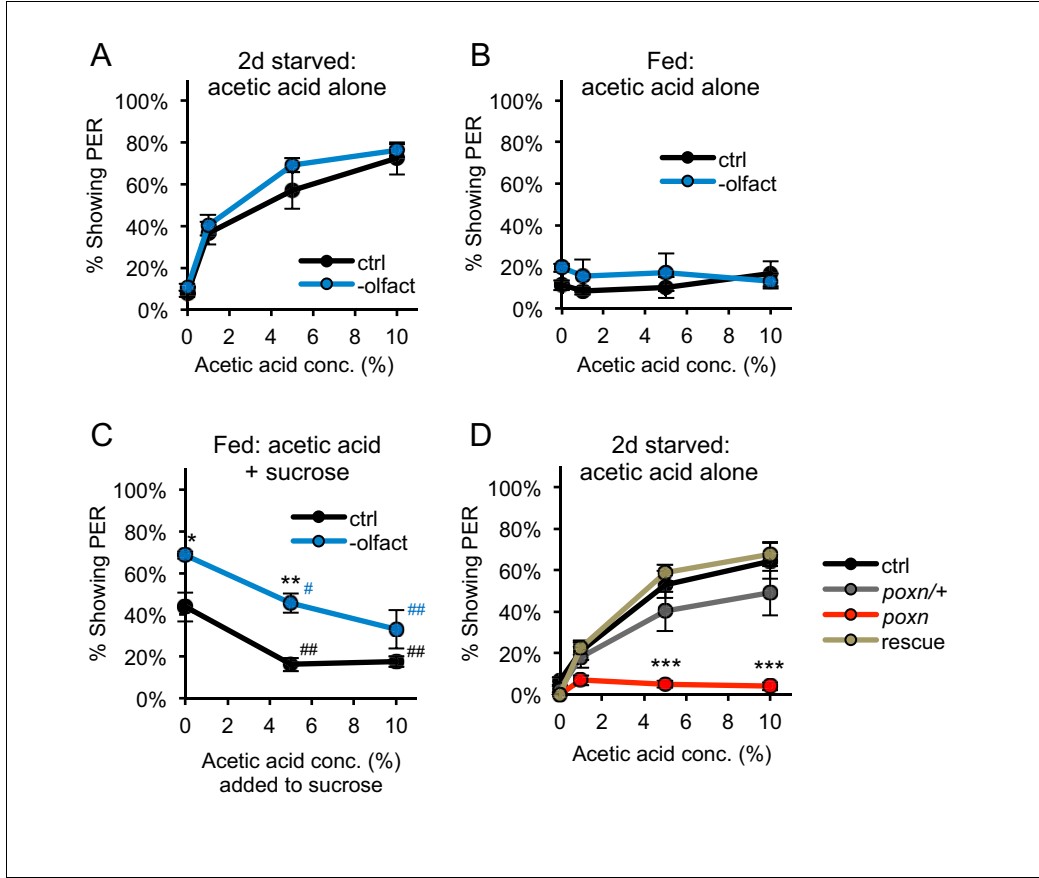

**Figure 2.** PER to acetic acid is mediated by the gustatory system, not the olfactory system. (**A–B**) Removing the olfactory organs did not affect PER to acetic acid in two-day starved flies (**A**) or fed flies (**B**) (p>0.05). (**C**) Acetic acid aversion in fed flies, measured by suppression of PER to 50 mM sucrose, was observed in both control flies and flies lacking olfactory organs. Flies lacking olfactory organs were generally more responsive than control flies. (**D**) Two-day starved flies homozygous for the $poxn^{\Delta M22-B5}$ mutation showed decreased PER to acetic acid as compared to wild-type controls, $poxn^{\Delta M22-B5}/+$ heterozygotes, and $poxn^{\Delta M22-B5}$ homozygotes carrying a rescue transgene. In all experiments different groups were compared by two-way repeated measures ANOVA followed by Bonferroni's post-tests (*p<0.05, **p<0.01, ***p<0.001). In panel C, responses within each group were compared to the response to 0% acetic acid using one-way repeated-measures ANOVA followed by Dunnett's post-tests (#p<0.05, ##p<0.01; symbols colored by group). n = 3–5 sets of flies.
DOI: https://doi.org/10.7554/eLife.47677.010

The following source data is available for figure 2:

**Source data 1.** Raw data for *Figure 2*.
DOI: https://doi.org/10.7554/eLife.47677.011

for PER to acetic acid. An RNAi transgene targeting the gene *norpA*, a fly ortholog of PLC, was expressed in sugar neurons under the control of *Gr64f-Gal4*. RNAi inhibition of *norpA* expression in the sugar neurons severely reduced PER to fatty acids whereas PER to either sucrose or acetic acid was unaffected (*Figure 3E–G*). These data suggest that the appetitive response to acetic acid is mediated by sugar-sensing neurons, but engages molecular pathways distinct from those employed in the detection of either sugars or fatty acids.

## Bitter-sensing neurons suppress PER to acetic acid

We next identified the neurons that mediate the aversive response to acetic acid in fed flies. Multiple classes of bitter sensory neurons reside in the labellum, and each of these neurons expresses the receptor *Gr66a* (*Weiss et al., 2011*). We therefore employed the regulatory sequences of *Gr66a* to drive the expression of *Kir2.1* to silence the bitter neurons. In initial experiments we demonstrated

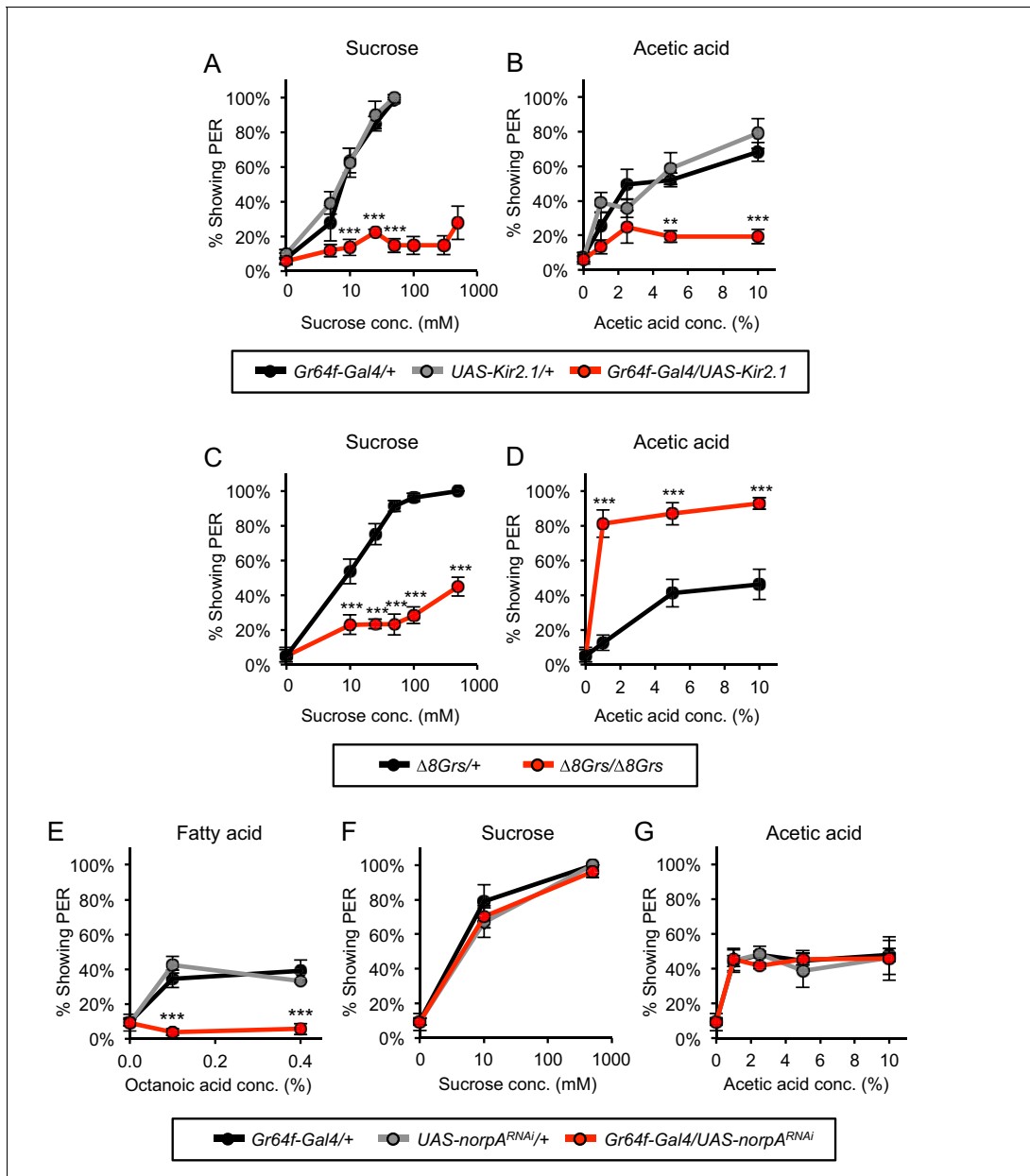

**Figure 3.** Sugar-sensing neurons mediate PER to acetic acid in starved flies. (**A–B**) Silencing the activity of sugar-sensing neurons impaired PER to sucrose (**A**) and acetic acid (**B**) in two-day starved flies. (**C–D**) One-day starved homozygous mutant flies lacking all eight sugar receptors showed decreased PER to sucrose (**C**) but showed increased PER to acetic acid (**D**) relative to heterozygous controls. (**E–G**) RNAi knockdown of *norpA*, encoding PLCß, in sugar-sensing neurons abolished PER to fatty acid (**E**) but did not affect PER to sucrose (**F**) or acetic acid (**G**) in two-day starved flies. For all panels: **p<0.01, ***p<0.001, two-way repeated measures ANOVA followed by Bonferroni's post-tests comparing experimental group to each control group. In panel A, statistical analyses did not include the three highest sucrose concentrations because control flies were not tested at these concentrations. n = 3–5 sets of flies.

DOI: https://doi.org/10.7554/eLife.47677.012

The following source data is available for figure 3:

**Source data 1.** Raw data for *Figure 3*.
DOI: https://doi.org/10.7554/eLife.47677.013

the efficacy of Kir2.1 silencing. Starved control flies exhibit PER to sucrose, and this response is strongly diminished by the addition of the bitter compounds quinine or lobeline (*Figure 4—figure supplement 1A–B*). This suppression of PER by bitter compounds is no longer observed when Kir2.1 is expressed in bitter neurons, demonstrating the efficacy of Kir2.1 silencing (*Figure 4—figure supplement 1A–B*). We therefore employed Kir2.1 to examine the effect of silencing bitter neurons on the aversive responses to acetic acid in fed flies. In control fed flies acetic acid suppressed PER to sucrose, but this suppression was largely eliminated when the bitter neurons were silenced (*Figure 4A*). In controls,~80% of flies exhibited PER to sucrose alone, and this was reduced to ~20–30% upon addition of 10% acetic acid. In contrast, when bitter neurons were silenced over 80% of flies continued to exhibit PER upon exposure to a mixture of sugar and 10% acetic acid (*Figure 4A*). These results indicate that bitter-sensing neurons mediate the aversive response to acetic acid in fed flies.

We also examined the consequences of bitter neuron silencing on the responses to acetic acid alone. Starved flies exhibit PER to acetic acid alone, but this response is not observed in fed flies (*Figure 1A*). Acetic acid elicited PER in ~20% of control fed flies, a value near baseline (*Figure 4B*). Silencing of the bitter neurons resulted in a striking increase in the percentage of fed flies that exhibited PER to acetic acid (~60–80%; *Figure 4B*). Silencing bitter neurons in fed flies did not affect PER to sucrose alone, indicating that the bitter neurons do not exert nonspecific suppression of PER (*Figure 4—figure supplement 1C*). These results afford an explanation for the observation that fed flies normally fail to exhibit PER to acetic acid. Our data suggest that acetic acid activates sugar-sensing neurons, which promote an appetitive response, but in the fed state simultaneous activation of bitter-sensing neurons completely suppresses this response. Silencing the bitter neurons eliminates this suppression, unmasking the appetitive response.

Silencing the bitter neurons resulted not only in the emergence of PER to acetic acid in fed flies but also enhanced PER to acetic acid in starved flies (*Figure 4C*). PER to 10% acetic acid was observed in 70–80% of control flies and this value increased to 96% upon bitter neuron silencing (*Figure 4C*). Silencing bitter neurons in starved flies did not affect PER to sucrose (*Figure 4—figure supplement 1D*). These results demonstrate that even in the starved state, bitter neurons suppress PER to acetic acid. The observation that bitter neuron silencing has a stronger effect on acetic acid-induced PER in fed flies (*Figure 4B*) than in starved flies (*Figure 4C*) suggests that activity within the bitter-sensing circuit is suppressed in the starved state. This inhibition by hunger could occur either within or downstream of bitter sensory neurons.

The striking increase in PER to acetic acid after starvation results from the hunger-dependent suppression of the bitter-sensing circuit but may also reflect enhancement of the appetitive sugar-sensing pathway. We and others observe that PER to sucrose is increased by starvation, indicating that the neural pathway for sugar-sensing is upregulated by hunger (*Figure 1B*; *Inagaki et al., 2012*). We therefore examined the relative contributions of the sugar- and bitter-sensing pathways to acetic acid-induced PER. We compared responses to acetic acid in both fed and starved flies with and without bitter neuron silencing. In control flies, starvation strongly increased PER to acetic acid:~20% of fed flies and 70–80% of starved flies responded at the highest concentration (*Figure 4D–E*). Upon bitter neuron silencing the difference between PER in fed and starved flies was still observed, but was much smaller in magnitude: 68% of fed flies and 96% of starved flies responded at the highest concentration (*Figure 4F*). This starvation-dependent enhancement of PER in bitter-silenced flies is likely to reflect the enhancement of the sugar-sensing circuit. These results reveal a state-dependent interaction between the bitter- and sugar-sensing circuits that affords a logic for the behavioral switch. In fed flies, bitter neurons strongly suppress the appetitive response to acetic acid mediated by sugar neurons. Hunger results in a behavioral switch that increases PER both by suppressing the bitter-sensing circuit and enhancing the sugar-sensing circuit.

## IR25a and IR76b are not required for acetic acid responses

Two members of the ionotropic receptor (IR) family of chemoreceptors, IR25a and IR76b, function in tarsal sugar-sensing neurons to detect fatty acids and in a separate population of tarsal sour-sensing neurons to detect organic and inorganic acids (*Ahn et al., 2017*; *Chen and Amrein, 2017*). These IRs also mediate salt detection in multiple classes of labellar taste neurons, including sugar-sensing neurons (*Zhang et al., 2013*; *Jaeger et al., 2018*). We therefore tested whether IR25a and IR76b are required for appetitive or aversive taste responses to acetic acid.

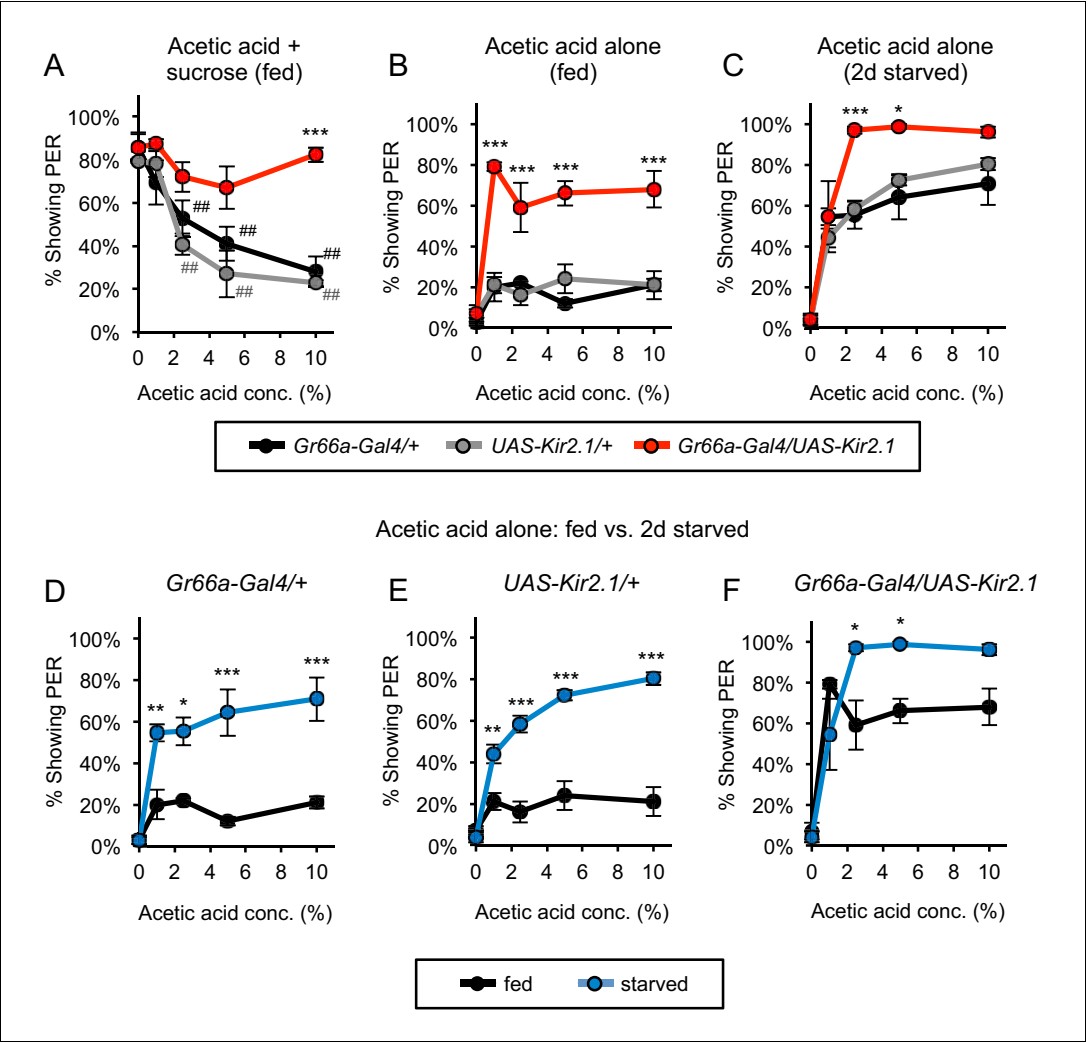

**Figure 4.** Bitter-sensing neurons suppress PER to acetic acid. (**A**) Silencing bitter-sensing neurons strongly reduced aversion to acetic acid in fed flies. Aversion was measured as the suppression of PER to 100 mM sucrose containing acetic acid. Both sets of control flies showed significant aversion to acetic acid (##$p < 0.01$; symbols colored by group), whereas experimental flies did not show significant aversion (one-way repeated-measures ANOVA followed by Dunnett's post-tests comparing responses to 0% acetic acid to responses at all other concentrations). (**B–C**) Silencing bitter-sensing neurons enhanced PER to acetic acid alone in fed flies (**B**) and two-day starved flies (**C**). (**D–F**) Comparing fed and starved flies of each genotype (same data as panels B and C) revealed that starvation enhanced PER in all genotypes. For all panels: *$p < 0.05$, **$p < 0.01$, ***$p < 0.001$, two-way ANOVA (repeated measures for panels A-C) followed by Bonferroni's post-tests comparing experimental group to each control group. n = 3–4 sets of flies. See also *Figure 4—figure supplements 1* and *2*.
DOI: https://doi.org/10.7554/eLife.47677.014

The following source data and figure supplements are available for figure 4:

**Source data 1.** Raw data for *Figure 4*.
DOI: https://doi.org/10.7554/eLife.47677.017

**Figure supplement 1.** Silencing bitter-sensing neurons impairs bitter aversion but does not affect PER to sugar.
DOI: https://doi.org/10.7554/eLife.47677.015

**Figure supplement 2.** Acetic acid responses in fed and starved flies carrying mutations in *IR25a* or *IR76b*.
DOI: https://doi.org/10.7554/eLife.47677.016

We tested the *Ir25a* and *IR76b* mutant lines that exhibit impairments in fatty acid and sour taste detection as well as the control strain (*w[1118]*) used in these studies (*Ahn et al., 2017*; *Chen and Amrein, 2017*). Starved flies carrying mutations in *Ir25a* or *Ir76b* showed robust PER to 1–10% acetic acid, indicating that these receptors are not required for the appetitive response to acetic acid (*Figure 4—figure supplement 2A and D*). Fed flies carrying *IR25a* or *IR76b* mutations showed very low levels of PER to acetic acid, indicating that the aversive pathway that suppresses acetic acid-evoked PER in the fed state remains intact (*Figure 4—figure supplement 2B and E*). We note that both fed and starved *IR25a* mutant flies showed slightly lower PER to acetic acid than controls, suggesting that IR25a may contribute to this response even though it is not strictly required for acetic acid detection. We also tested the ability of acetic acid to suppress sucrose-evoked PER in fed flies. In *IR25a* mutants, the suppression of sucrose-evoked PER was similar or stronger than we observed in control *w[1118]* flies (*Figure 4—figure supplement 2C*). *IR76b* mutants also showed a decrease in sucrose-evoked PER as the acetic acid concentration was increased from 1% to 10%, but low concentrations of acetic acid unexpectedly enhanced their response to sucrose (*Figure 4—figure supplement 2F*). This enhancement is likely due to the genetic background of the flies and not to the loss of IR76b because *IR76b/+* heterozygotes, which carry one functional copy of *IR76b*, showed the same effect, and their responses to sucrose containing acetic acid did not significantly differ from homozygotes (*Figure 4—figure supplement 2G–H*). Taken together, these experiments indicate that IR25a and IR76b are not required for either appetitive or aversive responses to acetic acid.

## Acetic acid activates sugar- and bitter-sensing neurons

The observation that the appetitive response to acetic acid is mediated by sugar-sensing neurons whereas the aversive response is mediated by bitter neurons suggests that acetic acid is an unusual tastant capable of activating two opposing classes of sensory cells. We therefore performed two-photon imaging of taste sensory neurons to confirm whether acetic acid activates both sugar- and bitter-sensing cells. The genetically encoded calcium indicator *GCaMP6f* (*Chen et al., 2013*) was expressed in sugar- (*Gr64f-Gal4*) or bitter- (*Gr66a-Gal4*) sensing neurons. Imaging was performed on sensory axon termini in the subesophageal zone (SEZ) of the fly brain (*Figure 5—figure supplement 1*). Strong GCaMP responses to acetic acid were observed in sugar neurons in both fed and starved flies, with the peak response to acetic acid about half of that observed with 500 mM sucrose (*Figure 5A–D*). We noted that the responses to acetic acid stimuli often appeared more variable across trials and flies than responses to sucrose (*Figure 5D*; *Figure 5—figure supplement 2*). Despite this variability, sugar neurons in 27 of 28 fed or starved flies responded to acetic acid at levels greater than the water response.

We also examined the acetic acid response in sugar neurons of fed homozygous mutant flies carrying deletions in all eight sugar receptors (*Δ8Grs/Δ8Grs*). The response to sucrose in these mutants was reduced to the level of the response to water, whereas the response to acetic acid was not affected (*Figure 5—figure supplement 3*). Acetic acid activation of sugar neurons was variable across flies, but the proportion of flies responding to acetic acid did not differ by genotype: sugar neurons in 6 of 9 control flies and 6 of 9 mutant flies showed responses to at least one concentration of acetic acid. Thus the response to acetic acid in sugar neurons does not require sugar receptors, a result consistent with the observation that sugar receptor mutants still show PER to acetic acid (*Figure 3D*). Acetic acid elicits stronger PER in these mutants than in controls but activates the sugar neurons to similar levels in fed flies. These results suggest that the enhanced PER of the mutants is not likely due to an accumulation of acetic acid receptors or increased acetic acid transduction in sugar-sensing neurons. Instead, the diminished response of sensory neurons to sugar as well as intensified hunger may lead to upregulation of the downstream sugar circuit and result in enhanced PER without changes in sensory neuron activation.

We next imaged the bitter-sensing neurons and observed significant GCaMP responses to both 1% and 5% acetic acid (*Figure 5E–H*). As in the sugar neurons, acetic acid responses in bitter neurons often appeared more variable across trials and flies than responses to bitter (*Figure 5H*; *Figure 5—figure supplement 2*), but neurons in the majority of both fed and starved flies responded to acetic acid (15/18 fed flies and 16/18 starved flies). Acetic acid activated the bitter neurons with peak responses about 30% of those obtained with 1 mM lobeline (*Figure 5E–H*). The difference between the levels of activity elicited by lobeline and acetic acid may reflect different sensitivities of bitter neurons to the two compounds or the activation of a smaller subset of neurons by acetic acid.

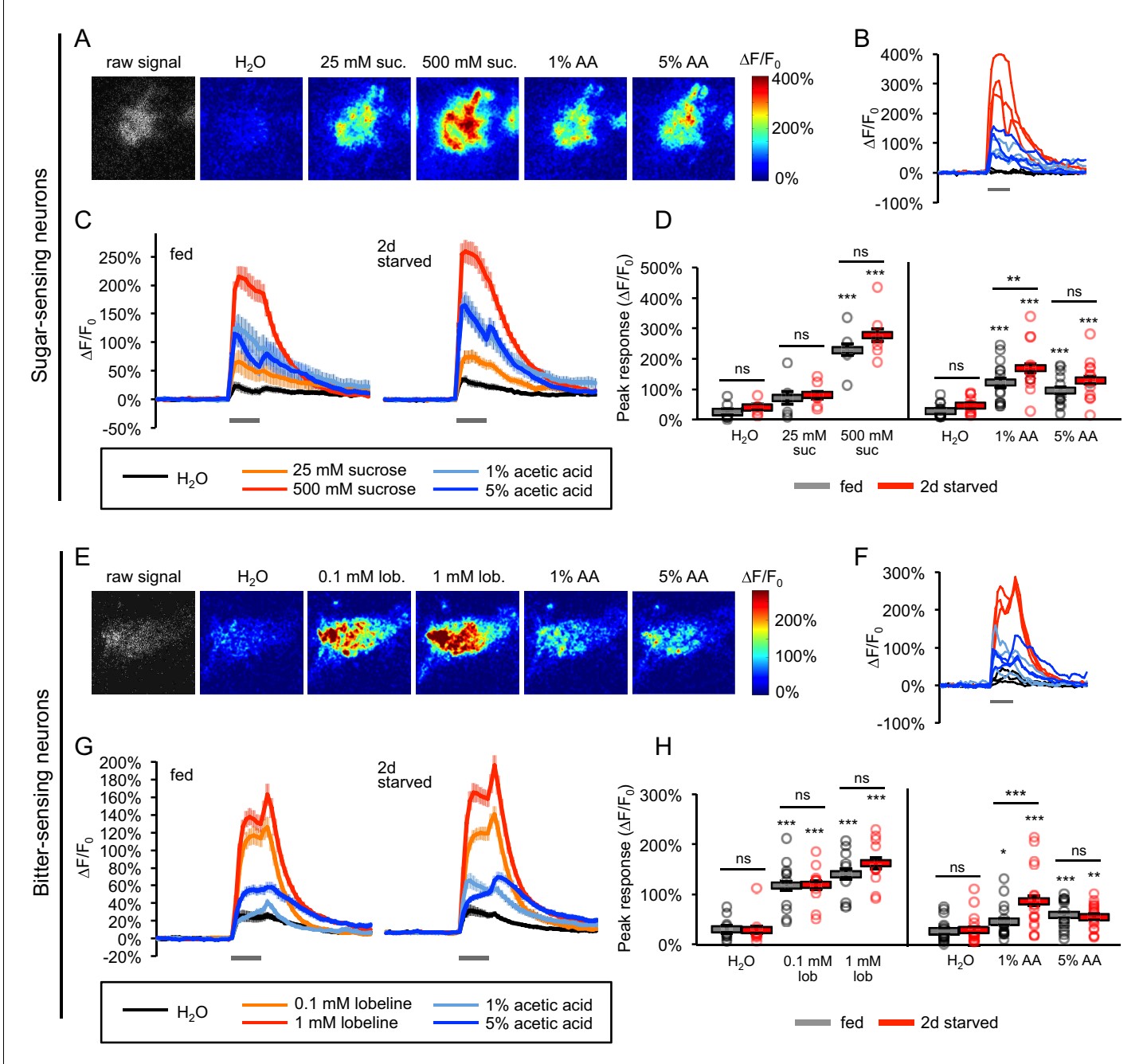

**Figure 5.** Acetic acid activates sugar- and bitter-sensing neurons. (A–H) Calcium imaging of taste sensory neurons reveals that acetic acid (AA) activates sugar-sensing neurons (labeled with *Gr64f-Gal4*; panels A-D) and bitter-sensing neurons (labeled with *Gr66a-Gal4*; panels E-H) in both fed and two-day starved flies. (A, E) Spatial maps of GCaMP activation by each stimulus for individual flies (A, fed; E, starved). (B, F) Example $\Delta F/F_0$ traces for individual trials in the same flies shown in A and E, respectively. (C, G) Average GCaMP activation across all trials in all flies of each group. Gray bars indicate stimulus delivery (2 s). (D, H) Peak response to each stimulus averaged across all trials for each group. Circles represent individual fly averages. Within each group, responses to each stimulus were compared to responses to water, and fed and starved groups were also compared for each stimulus (*p<0.05, **p<0.01, ***p<0.001, two-way ANOVA followed by Bonferroni post-tests). Data shown in this figure represent n = 15–19 trials, six flies per group (sugar neurons, sucrose stimuli), n = 39–43 trials, 14 flies per group (sugar neurons, AA stimuli), n = 30 trials, 10 flies per group (bitter neurons, lobeline stimuli), or n = 54 trials, 18 flies (bitter neurons, AA stimuli). n values are larger for AA stimuli because we combined the results of two different datasets, but only the more recent dataset included the appropriate sucrose and lobeline stimuli. Only the more recent dataset is shown in panels C and G, whereas peak responses to AA stimuli for the combined dataset are analyzed in panels D and H. See also *Figure 5—figure supplements 1–7*.
DOI: https://doi.org/10.7554/eLife.47677.018

The following source data and figure supplements are available for figure 5:

*Figure 5 continued on next page*

*Figure 5 continued*

**Source data 1.** Raw data for *Figure 5*.
DOI: https://doi.org/10.7554/eLife.47677.026
**Figure supplement 1.** Calcium imaging setup for taste neuron imaging.
DOI: https://doi.org/10.7554/eLife.47677.019
**Figure supplement 2.** GCaMP responses in individual flies.
DOI: https://doi.org/10.7554/eLife.47677.020
**Figure supplement 3.** Acetic acid responses of sugar-sensing neurons in sugar receptor mutants are not affected.
DOI: https://doi.org/10.7554/eLife.47677.021
**Figure supplement 4.** Acetic acid activates a subset of bitter-sensing neurons.
DOI: https://doi.org/10.7554/eLife.47677.022
**Figure supplement 5.** Responses of sugar- and bitter-sensing neurons to additional acetic acid concentrations.
DOI: https://doi.org/10.7554/eLife.47677.023
**Figure supplement 6.** Water-sensing neurons are activated by acetic acid only in accordance with its osmolarity.
DOI: https://doi.org/10.7554/eLife.47677.024
**Figure supplement 7.** GCaMP-expressing flies show hunger-dependent changes in PER to acetic acid and sucrose.
DOI: https://doi.org/10.7554/eLife.47677.025

We imaged the acetic acid response of four different subclasses of bitter neurons (*Weiss et al., 2011*) and observed that only the S-b class exhibited a significant peak response when compared to the response to water (*Figure 5—figure supplement 4*). This finding is consistent with studies showing the strongest acetic acid responses in bitter cells of S-type sensilla (*Rimal et al., 2019*) and specifically the S-b class (*Charlu et al., 2013*). We do not rule out the possibility that other classes of bitter neurons also respond to acetic acid (see *Figure 5—figure supplement 4*). Overall, these results show that acetic acid activates both sugar- and bitter-sensing taste neurons.

In both sugar- and bitter-sensing neurons, the average response magnitudes to 5% acetic acid were similar or slightly lower than the response to 1% (*Figure 5*). We therefore tested a broader range of acetic acid concentrations. Responses to higher concentrations such as 10% could not be accurately quantified because they frequently activated sensory neurons even before making contact with the labellum, likely due to volatile acetic acid molecules, so we focused on lower concentrations. In sugar neurons, we observed dose-dependent responses as the concentration was increased from 0.01% to 1%, but the response diminished slightly at 5% (*Figure 5—figure supplement 5A*). Bitter neurons failed to show dose dependence in this concentration range (*Figure 5—figure supplement 5B*). A diminished response over trials may contribute to the apparent lack of dose dependence, since we always tested acetic acid concentrations in ascending order. Indeed, testing 5% acetic acid prior to any other concentrations induced a much higher bitter neuron response ($113 \pm 15\%$ $\Delta F/F_0$, n = 18 trials, six flies) than in other experiments where 5% acetic acid was tested last (54–56% $\Delta F/F_0$; *Figure 5H* and *Figure 5—figure supplement 5B*). Overall, we find that bitter neurons fail to show clear dose-dependent responses to acetic acid and sugar neurons show dose dependence only at low concentrations. These results suggest that acetic acid may exert a more complex effect on sensory neurons than other tastants. For example, secondary effects on neuronal activity could be induced by low pH or by undissociated acetic acid molecules, which may cross the cell membrane and directly affect intracellular pathways (*DeSimone et al., 2001*; *Liman et al., 2014*).

Because acetic acid activates both sugar- and bitter-sensing neurons, we confirmed that acetic acid does not promiscuously activate all classes of taste neurons by imaging responses of water-sensing neurons, sensory cells in the labellum that respond to low osmolarity tastants (*Cameron et al., 2010*). Water-sensing neurons responded broadly to several taste stimuli at levels that were inversely related to their osmolarity (*Figure 5—figure supplement 6*). When the low-osmolarity response of water-sensing neurons was blocked by adding the high molecular mass polymer polyethylene glycol (PEG) to each taste solution, acetic acid did not activate the water-sensing neurons beyond the level elicited by PEG alone (*Figure 5—figure supplement 6C–D*). Acetic acid therefore activates water-sensing neurons solely by the osmolarity-sensing mechanism. Thus, the responses to acetic acid in sugar- and bitter-sensing neurons are specific and are likely to be mediated by receptors that recognize acetic acid.

## Hunger modulation of taste sensory neuron responses

We next compared the responses of sugar- and bitter-sensing neurons in fed and two-day starved flies. We confirmed that the GCaMP6f-expressing flies used for imaging show behavioral changes in response to hunger: like wild-type flies, they show strong hunger-dependent increases in PER to both acetic acid and sucrose (*Figure 5—figure supplement 7*). Previous studies suggest that hunger increases sugar neuron responses to sucrose and suppresses bitter neuron responses to lobeline (*Inagaki et al., 2012*; *Inagaki et al., 2014*; *LeDue et al., 2016*). We observed a trend toward increased sucrose responses in sugar neurons after starvation (p=0.063 for the effect of starvation, two-way ANOVA), but lobeline responses of bitter neurons did not differ between fed and starved flies (*Figure 5C–D and G–H*). Because *Inagaki et al. (2012)* detected differences between sucrose responses of fed and starved flies by quantifying the integrated $\Delta F/F_0$ response rather than the peak response, we also quantified integrated $\Delta F/F_0$ responses to sucrose, which showed a significant difference between fed and starved flies at 500 mM sucrose (p<0.05, two-way ANOVA followed by Bonferroni post-tests). Analyzing the integrated $\Delta F/F_0$ response did not reveal any significant differences between lobeline responses of bitter neurons in fed and starved flies.

We then examined sensory neuron responses to acetic acid in fed and starved flies. In both sugar and bitter neurons, starved flies showed significantly higher responses than fed flies to 1% but not 5% acetic acid (*Figure 5C–D and G–H*). Hunger increased the average peak response to 1% acetic acid from 117% to 166% $\Delta F/F_0$ in sugar neurons and from 46% to 87% $\Delta F/F_0$ in bitter neurons. In sugar neurons, but not bitter neurons, starved flies also showed a trend toward higher responses to 5% acetic acid than fed flies. Our genetic silencing data suggest that hunger elicits a behavioral switch in the acetic acid response by upregulating the sugar-sensing circuit and downregulating the bitter-sensing circuit. Thus the enhancement of the sugar neuron response to acetic acid in starved flies may contribute to the increase in PER, but the enhancement of the bitter neuron response is not consistent with the behavioral change. This effect on bitter neurons along with the lack of significant hunger-dependent changes at 5% acetic acid in either sugar or bitter neurons, despite the fact that PER to 5% acetic acid shows even greater hunger modulation than at 1% (*Figure 1A*; *Figure 5—figure supplement 7*), suggests that sensory neuron modulation is not likely to account for the behavioral switch in the acetic acid response. The striking effects of internal state on this behavior may therefore reflect state-dependent modulation of both the sugar and bitter circuits downstream of the sensory neurons.

## Discussion

Innate behaviors are observed in naive animals without prior learning or experience, suggesting that they are mediated by neural circuits that are genetically determined. Meaningful stimuli such as the taste and smell of food elicit stereotyped behaviors that are observed in all individuals in a species. However, innate behavioral responses can also exhibit flexibility and can be modulated by experience, expectation, and internal state (*Bargmann, 2012*; *Ding and Perkel, 2014*; *Kim et al., 2017*). Taste circuits in flies and mice appear to be anatomically and functionally programmed to elicit an innate appetitive or aversive response, ensuring that animals consume nutritious foods and avoid the ingestion of toxins (*Liman et al., 2014*). We have examined the response of *Drosophila* to acetic acid, a tastant that can be a metabolic resource but can also be toxic to the fly. Our data reveal that flies accommodate these conflicting attributes of acetic acid by virtue of a hunger-dependent switch in their behavioral response to this stimulus. Fed flies show taste aversion to acetic acid, likely a response to its potential toxicity, whereas starved flies show a robust appetitive response that may reflect their overriding need for calories. These opposing responses are mediated by two different classes of taste neurons. Acetic acid activates both the sugar and bitter pathways, which have opposing effects on feeding behavior. The choice of behaviors is determined by internal state: hunger shifts the response from aversion to attraction by enhancing the appetitive sugar pathway as well as suppressing the aversive bitter pathway (*Figure 6*).

## Activation of sugar and bitter neurons by acetic acid

The adaptive response to acetic acid is dependent on two biological features, the ability of acetic acid to activate two classes of sensory neurons that elicit opposing behaviors and the state-

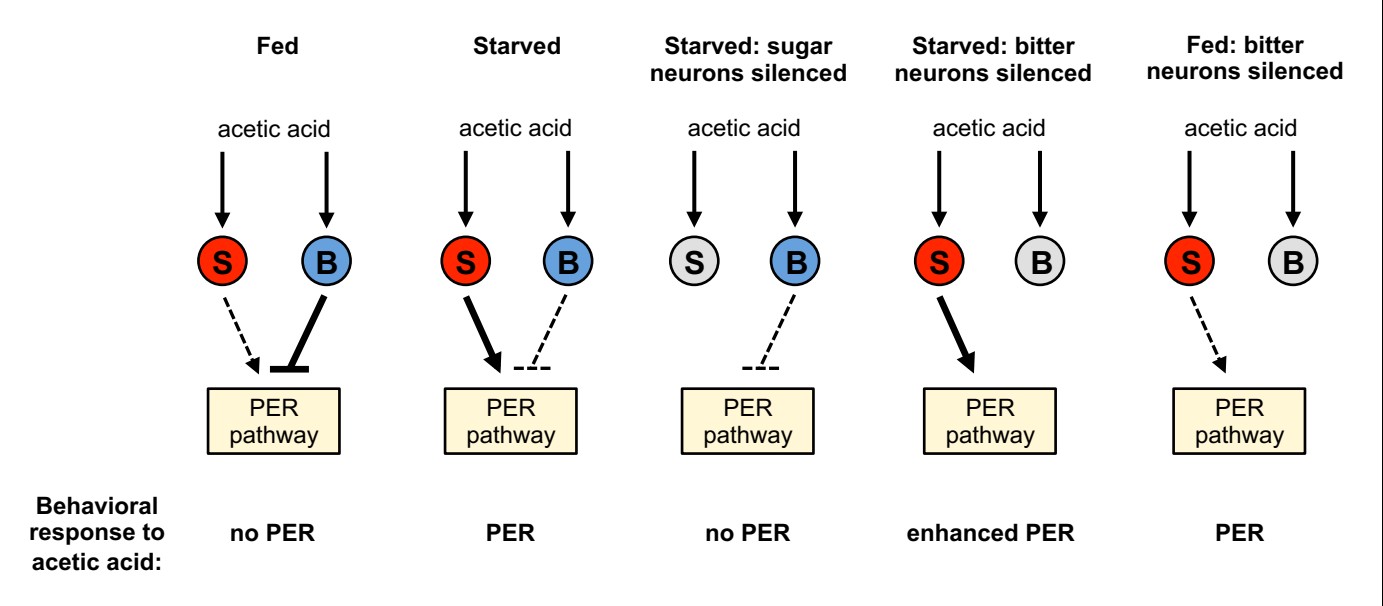

**Figure 6.** Model for a hunger-dependent switch in the behavioral response to acetic acid. Acetic acid activates both sugar- and bitter-sensing neurons ('S' and 'B' respectively). Sugar-sensing neurons promote PER to acetic acid whereas bitter-sensing neurons suppress PER. The balance of these two pathways determines the behavioral response. Hunger both enhances the sugar pathway and suppresses the bitter pathway, most likely by primarily acting downstream of sensory neurons. Thus the bitter pathway dominates in the fed state to suppress PER and elicit aversion, whereas the sugar pathway dominates in the starved state to elicit appetitive PER behavior. Silencing the sugar or bitter neurons (gray) shifts the balance of the two pathways to alter the behavioral response, as shown in the three right schematics.

DOI: https://doi.org/10.7554/eLife.47677.027

dependent modulation of these two taste pathways. We observe that acetic acid activates both the bitter- and sugar-sensing neurons. The activation of bitter neurons by organic acids has been observed previously by electrophysiologic recording of labellar sensilla, and these acids result in taste aversion (*Charlu et al., 2013*; *Rimal et al., 2019*). The observation that bitter neurons respond not only to organic acids but also to hydrochloric acid suggested that a subset of bitter neurons serve as a pH sensor eliciting taste aversion (*Charlu et al., 2013*). However, *Rimal et al. (2019)* recently identified IR7a as a narrowly tuned acetic acid receptor acting in bitter neurons. We have not tested whether IR7a mediates the aversive responses to acetic acid that we observe.

We also observe that acetic acid activates sugar neurons, eliciting an appetitive taste response. This behavioral response is not observed upon exposure to low pH or acetate, suggesting the presence of a receptor on sugar neurons recognizing short chain aliphatic acids. Two studies have reported that whereas acetic acid activates bitter neurons, it fails to activate sugar neurons (*Charlu et al., 2013*; *Rimal et al., 2019*). In these studies acetic acid activated S-type sensilla, which contain both bitter and sugar neurons, but failed to activate L-type sensilla, which contain sugar neurons but not bitter neurons. These results obtained by sensillar recordings contrast with our observations that acetic acid activates the axon termini of sugar neurons. One possible explanation for this discrepancy is that sugar neurons in S-type sensilla respond to acetic acid. *Rimal et al. (2019)* observe that in mutants lacking IR7a, S-type sensilla continue to exhibit weak responses to acetic acid, which may represent responses of sugar neurons. We are imaging axonal activity across the entire population of ~62 labellar sugar neurons, whereas electrophysiologic studies record single cells. Weaker dendritic responses not detected by extracellular sensilla recordings may be summated to produce observable axonal responses. In addition, nonlinear amplification of spike rates into calcium responses or presynaptic facilitation by modulatory inputs could also transform weak dendritic responses into stronger axonal signals.

We are confident in our result that sugar neurons show axonal GCaMP responses to acetic acid. Acetic acid activated sugar neurons in 27 of 28 flies initially tested (*Figure 5A–D*), and overall we have observed acetic acid activation of sugar neurons with three different *Gal4* drivers and two

different GCaMP variants (*Figure 5A–D*, *Figure 5—figure supplement 3*, and data not shown). Moreover, the observation that acetic acid elicits PER in starved flies and this response is eliminated upon silencing activity of the sugar neurons is most consistent with the fact that sugar neurons are activated by acetic acid. It is puzzling, however, that the dose-dependence of sugar neuron responses does not parallel the dose-dependence of acetic acid-evoked PER, which increases from 1% to 10% acetic acid (*Figure 1A*; *Figure 5—figure supplement 7*). One possibility is that a subset of sugar neurons are activated dose-dependently and contributes most strongly to PER, and we may not observe this dose-dependence when imaging activity across all sugar neurons. Alternatively, despite our finding that sugar neuron activity is required for PER to acetic acid, additional sensory neurons may contribute to this behavior and detect acetic acid in a dose-dependent manner.

Silencing sugar or bitter neurons largely abolishes the appetitive or aversive response to acetic acid, respectively, suggesting that we have identified the primary neurons that mediate these behaviors. However, our experiments do not rule out the possibility that other labellar taste neurons contribute to these behaviors. The fact that the leg contains dedicated acid-sensing neurons raises the possibility of whether such neurons also exist in the labellum (*Chen and Amrein, 2017*). The tarsal acid-sensing neurons utilize IR76b and IR25a, which are dispensable for the acetic acid-induced behaviors we have studied (*Figure 4—figure supplement 2*), suggesting that putative labellar acid-sensing neurons would utilize a different molecular mechanism.

The activation of different classes of sensory neurons by a single tastant has been observed for salt (*Zhang et al., 2013*; *Jaeger et al., 2018*), the long chain fatty acid, hexanoic acid (*Ahn et al., 2017*), as well as for acetic acid in this study. In both flies and mammals, low salt concentrations elicit attraction whereas high salt results in aversion and these opposing behaviors are mediated by distinct classes of sensory neurons (*Zhang et al., 2013*; *Jaeger et al., 2018*; *Chandrashekar et al., 2010*; *Oka et al., 2013*). Hexanoic acid, a caloric source, activates fatty acid receptors in sugar neurons and at high concentrations activates a different receptor in bitter neurons (*Ahn et al., 2017*). A logical pattern emerges in which tastants of potential value to the fly activate attractive taste pathways. These compounds may also be toxic and also activate aversive pathways either at higher concentrations or in different internal states. This affords the fly protection from the potential toxicity of excess, a protection that can be ignored under extreme conditions to assure survival.

## Hunger modulation of neural circuits and behavior

Internal state can elicit profound behavioral changes that allow the organism to adapt to a changing internal world. Hunger, for example, results in enhanced food search and consumption, increased locomotion, changes in food preference, and altered olfactory and taste sensitivity (*Sternson et al., 2013*; *Itskov and Ribeiro, 2013*; *Pool and Scott, 2014*; *Yang et al., 2015*). Previous studies have observed that hunger enhances olfactory attraction to cider vinegar, increases sugar sensitivity, and decreases bitter sensitivity (*Root et al., 2011*; *Inagaki et al., 2012*; *Inagaki et al., 2014*). These effects of hunger represent gain changes; stimuli become more attractive or aversive. By contrast, our experiments reveal a qualitative change in the valence of acetic acid: hunger induces a switch from taste aversion to attraction.

Genetic silencing experiments indicate that this switch in starved flies results from enhancement of the appetitive pathway, mediated by sugar neurons, and inhibition of the aversive pathway mediated by bitter neurons. Conversely, in fed flies the appetitive pathway is inhibited whereas the aversive pathway is enhanced. Previous studies suggest that hunger modulates behavioral responses to sugar and bitter at least in part by modulating the activity of the initial neurons in these pathways, the sensory neurons (*Inagaki et al., 2012*; *Inagaki et al., 2014*; *LeDue et al., 2016*). Our experiments imaging sensory neuron projections in the SEZ revealed a trend toward increased sugar neuron responses to sucrose in starved flies but did not show hunger modulation of the bitter neuron response to lobeline. It is possible that hunger modulation would have been apparent if we tested a greater range of concentrations or that fly to fly variability precluded us from detecting subtle differences. We did observe that hunger significantly increased the responses of both sugar and bitter neurons to 1% acetic acid, and sugar neurons also showed a trend toward enhanced responses at 5%. Thus the enhancement of sugar neuron responses may contribute to increased acetic acid-evoked PER in starved flies. However, modulation of sensory neurons is unlikely to entirely account for the behavioral switch in the acetic acid response for multiple reasons. First, behavioral data suggest that hunger suppresses the bitter circuit, but bitter-sensing neurons show an enhanced

response to acetic acid. Second, flies show a greater hunger-dependent change in PER to 5% than 1% acetic acid, but sensory neuron responses only show significant modulation at 1%.

Our data therefore suggest that the striking state-dependent switch in the behavioral response to acetic acid may reflect modulation of taste pathways downstream of the sensory neurons. The circuit from sensory neurons leading to proboscis extension remains largely uncharacterized but multiple nodes subject to modulation can be anticipated. The response to sugar extends to multiple behaviors beyond PER including ingestion, swallowing, and suppression of locomotion, each of which is likely to be modulated by hunger (*Pool and Scott, 2014*). Modulation at the level of sensory neurons affords gain control that will result in changes in all behaviors elicited by gustatory neurons. Modulation of downstream taste neurons facilitating sensorimotor transformations could afford a flexibility enabling independent control of different behavioral programs driven by the same taste stimulus. This affords genetically determined neural circuits mediating innate behaviors the opportunity for more complex modulation dependent on perception, motivation, and internal state.

# Materials and methods

**Key resources table**

| Reagent type (species) or resource | Designation | Source or reference | Identifiers | Additional information |
|---|---|---|---|---|
| Genetic reagent (*Drosophila melanogaster*) | wild-type control 2U (isoCJ1) | *Dubnau et al., 2001* | | |
| Genetic reagent (*D. melanogaster*) | poxn[ΔM22-B5] | *Boll and Noll, 2002* | Flybase: FBal0144686 | |
| Genetic reagent (*D. melanogaster*) | poxn[ΔM22-B5] with *SuperA* (rescue) | *Boll and Noll, 2002* | Flybase: FBal0144670 | |
| Genetic reagent (*D. melanogaster*) | Δ8Grs (R1, ΔGr5a;; ΔGr61a, ΔGr64a-f) | *Yavuz et al., 2014* | | |
| Genetic reagent (*D. melanogaster*) | R1, ΔGr5a; Gr61a-Gal4, UAS-GCaMP6m; ΔGr61a, ΔGr64a-f | *Yavuz et al., 2014* | | |
| Genetic reagent (*D. melanogaster*) | IR25a[1] | *Benton et al., 2009* | Flybase: FBst0041736 | |
| Genetic reagent (*D. melanogaster*) | IR25a[2] | *Benton et al., 2009* | Flybase: FBst0041737 | |
| Genetic reagent (*D. melanogaster*) | IR76b[1] | *Zhang et al., 2013* | Flybase: FBst0051309 | |
| Genetic reagent (*D. melanogaster*) | IR76b[2] | *Zhang et al., 2013* | Flybase: FBst0051310 | |
| Genetic reagent (*D. melanogaster*) | w[1118] | Amrein lab | Flybase: FBst0003605 | |
| Genetic reagent (*D. melanogaster*) | Gr64f-Gal4 | *Dahanukar et al., 2007* | Flybase: FBtp0057275 | |
| Genetic reagent (*D. melanogaster*) | Gr66a-Gal4 | *Scott et al., 2001* | Flybase: FBtp0014661 | |
| Genetic reagent (*D. melanogaster*) | Gr98d-Gal4 | *Weiss et al., 2011* | Flybase: FBst0057692 | |
| Genetic reagent (*D. melanogaster*) | Gr22f-Gal4 | *Weiss et al., 2011* | Flybase: FBst0057610 | |
| Genetic reagent (*D. melanogaster*) | Gr59c-Gal4 | *Weiss et al., 2011* | Flybase: FBst0057650 | |
| Genetic reagent (*D. melanogaster*) | Gr47a-Gal4 | *Weiss et al., 2011* | Flybase: FBst0057638 | |

*Continued on next page*

*Continued*

| Reagent type (species) or resource | Designation | Source or reference | Identifiers | Additional information |
|---|---|---|---|---|
| Genetic reagent (*D. melanogaster*) | *ppk28-Gal4* | *Cameron et al., 2010* | Flybase: FBtp0054514 | |
| Genetic reagent (*D. melanogaster*) | *UAS-Kir2.1* | *Baines et al., 2001* | Flybase: FBtp0014166 | |
| Genetic reagent (*D. melanogaster*) | *UAS-GCaMP6f* | *Chen et al., 2013* | Flybase: FBst0042747 | |
| Genetic reagent (*D. melanogaster*) | *UAS-norpA$^{RNAi}$* | *Masek and Keene, 2013* | Flybase: FBst0031113 | |
| Chemical compound, drug | acetic acid | Sigma-Aldrich | 338826 | |
| Chemical compound, drug | sucrose | Sigma-Aldrich | S9378 | |
| Chemical compound, drug | lobeline hydrochloride | Sigma-Aldrich | 141879 | |
| Chemical compound, drug | quinine hydrochloride dihydrate | Sigma-Aldrich | Q1125 | |
| Chemical compound, drug | myristic acid | Sigma-Aldrich | M3128 | |
| Software, algorithm | Prism, version 4 | GraphPad | | |
| Software, algorithm | MATLAB | Mathworks | | |
| Other | two-photon laser scanning microscope | Ultima, Bruker | | |
| Other | Ti:S laser | Chameleon Vision, Coherent | | |
| Other | GaAsP detector | Hamamatsu Photonics | | |

## Fly stocks and maintenance

Flies were reared at 25°C and 70% relative humidity on standard cornmeal food. The wild-type control strain was *2U* (*isoCJ1*; *Dubnau et al., 2001*). All lines used for behavior were outcrossed into this background for at least five generations, with the exception of the *Δ8Grs* line which contained too many mutations to outcross and the *IR25a* and *IR76b* mutants which were tested with the $w^{1118}$ controls that other studies have used (*Chen and Amrein, 2017*; *Ahn et al., 2017*). PER assays were generally performed on 3–6 day-old mated females. Calcium imaging was performed on >1 week-old flies to ensure robust GCaMP6f expression, and PER assays for GCaMP6f-expressing flies were performed using flies of the same age.

All fly strains have been described previously: *Gr64f-Gal4* (*Dahanukar et al., 2007*); *Gr66a-Gal4* (*Scott et al., 2001*); *ppk28-Gal4* (*Cameron et al., 2010*); *Gr98d-Gal4*, *Gr22f-Gal4*, *Gr59c-Gal4*, and *Gr47a-Gal4* (*Weiss et al., 2011*); *UAS-Kir2.1* (*Baines et al., 2001*); *UAS-GCaMP6f* (*Chen et al., 2013*); *UAS-norpA$^{RNAi}$* (*Masek and Keene, 2013*); *poxn$^{ΔM22-B5}$* and *poxn$^{ΔM22-B5}$ + SuperA* rescue (*Boll and Noll, 2002*); *Δ8Grs* (R1, *ΔGr5a;; ΔGr61a, ΔGr64a-f*) and *Δ8Grs* with transgenes for GCaMP imaging (R1, *ΔGr5a; Gr61a-Gal4, UAS-GCaMP6m; ΔGr61a, ΔGr64a-f*) (*Yavuz et al., 2014*); *IR25a$^1$* and *IR25a$^2$* (*Benton et al., 2009*); *IR76b$^1$* and *IR76b$^2$* (*Zhang et al., 2013*).

## PER assay

Fed flies were taken directly from food vials for testing. Starved flies were food-deprived with water (using a wet piece of Kimwipe) for the specified amount of time before testing. Flies were anesthetized on ice and immobilized on their backs with myristic acid. Unless otherwise specified, PER experiments were conducted by taste stimulation of the labellum. To ensure that we could deliver

tastants to the labellum without contacting the legs, we immobilized the two anterior pairs of legs with myristic acid. For leg stimulation experiments (*Figure 1E–F*, *Video 1*, and *Video 2*), all legs remained free. Flies recovered from gluing for 30–60 min in a humidified chamber before testing.

Before testing PER, flies were water-satiated so that thirst would not affect their responses. PER to water (the negative control) was tested after water-satiation, followed by taste stimuli in ascending order of concentration. Flies were water-satiated again before each test. Each test consisted of two trials in which the solution was briefly applied to the labellum or legs using a small piece of Kimwipe. PER on at least one of the two trials was considered a positive response. Only full proboscis extensions, not partial extensions, were counted as PER. Flies were tested in groups of 15–20, and the percent of flies showing PER to each tastant was manually observed and recorded. Flies that did not respond to any taste stimuli were tested with 500 mM sucrose at the end of the assay. For experiments using only wild-type starved flies, which should always respond to high concentrations of sugar unless they are extremely unhealthy, flies that failed to respond to 500 mM sucrose were excluded from analysis. For experiments comparing fed and starved flies or starved controls and mutants, flies were only excluded from analysis if they appeared very sick.

For statistical analyses of PER, each group of 15–20 flies was considered to be a single data point ('n'). A minimum of three groups per genotype or condition were tested for each PER experiment. Because PER can vary substantially from day to day (possibly due to changes in ambient temperature or humidity), control and experimental flies for a given experiment were always tested on the same days, and all experiments were repeated over multiple days.

To test directional PER, we contacted the left or right forelegs with acetic acid, alternating between sides every 1–2 trials. Flies were filmed and the videos were analyzed later. We only analyzed trials in which flies showed full PER to the stimulus. Flies often showed repeated extension to a single stimulation; at least one proboscis extension toward the left or right side was considered to be a lateralized response.

To test the role of olfaction, the third antennal segments and maxillary palps were removed with forceps while flies were anesthetized on ice. Surgery was performed prior to starvation, and after surgery flies were given ~30 min to recover in food vials before starvation. Control flies were anesthetized for the same duration as antennectomized flies.

## Calcium imaging

Flies for calcium imaging were taped on their backs to a piece of clear tape in an imaging chamber (see *Figure 5—figure supplement 1*). Fine strands of tape were used to restrain the legs, secure the head, and immobilize the proboscis in an extended position for tastant stimulation. A small hole was cut into the tape to expose the anterior surface of the fly's head. A square hole along the anterior surface of the head was then cut through the cuticle, including removal of the antennae, to expose the anterior ventral aspect of the brain that encompasses the SEZ. The esophagus was cut in order to visualize the SEZ clearly. The dissection and imaging were performed in modified artificial hemolymph in which 15 mM ribose is substituted for sucrose and trehalose (*Wang et al., 2003*; *Marella et al., 2006*).

Calcium imaging experiments were performed using a two-photon laser scanning microscope (Ultima, Bruker) equipped with an ultra-fast Ti:S laser (Chameleon Vision, Coherent) that is modulated by pockel cells (Conoptics). Emitted photons were collected with a GaAsP photodiode detector (Hamamatsu) through a 60X water-immersion objective (Olympus). A single plane through the brightest area of axonal projections was chosen for imaging. Images were acquired at 925 nm at a resolution of 256 by 256 pixels and a scanning rate of 3–4 Hz.

Tastants were delivered to the labellum via a custom-built solenoid pinch valve system controlled by MATLAB software. Pinch valves were opened briefly (~10 ms) to create a small liquid drop at the end of a 5 µL glass capillary, positioned such that the drop would make contact with the labellum. Tastants were removed after a fixed duration by a vacuum line controlled by a solenoid pinch valve. Proper taste delivery was monitored using a side-mounted camera (Veho VMS-004), which allowed for visualization of the fly and tastant capillary using the light from the imaging laser. At least three trials of each stimulus were given, with at least one minute rest between trials to avoid habituation.

Calcium imaging data were analyzed using custom MATLAB code based largely on the code used in *Hattori et al. (2017)*. Images were registered within and across trials to correct for movement in the x-y plane using a sub-pixel registration algorithm (*Guizar-Sicairos et al., 2008*). Regions

of interest (ROIs) were drawn manually around the area of axonal projections. Average pixel intensity within the ROI was calculated for each frame. The average signal for 20 frames preceding stimulus delivery was used as the baseline signal ($F_0$), and the $\Delta F/F_0$ values for each frame were then calculated. The peak stimulus response was quantified as the average of the $\Delta F/F_0$ values for the two highest consecutive frames during tastant presentation. No trials were excluded from analysis unless the tastant drop failed to make proper contact with the labellum. For fly by fly analyses, we defined a fly as responding to a tastant if the average peak response across at least three trials was higher than the average peak response to water by a magnitude of at least 15%. We also considered thresholds of 10% or 20% but found that 15% appeared to be a reasonable (and likely conservative) threshold for defining a fly's response.

## Statistical analyses

Statistical analyses were performed using GraphPad Prism, Version 4. The most relevant statistical results are reported in the figures and legends, and all statistical results are reported in *Supplementary file 1*. All graphs represent mean ± SEM. For *Gal4/UAS* experiments, statistical significance was attributed only to data points for which experimental flies that differed from both the *Gal4/+* and *UAS/+* controls in the same direction. Sample sizes are listed in the figure legends. No explicit power analyses were used to determine sample sizes prior to experimentation. Minimum sample sizes were decided prior to experimentation based on previous experience knowing how many samples are usually sufficient to detect reasonable effect sizes. Additional samples were added if the initial results were inconclusive or more variable than expected, but never with the intent to make a non-significant p-value significant or vice versa. For experiments in which the same genotype was tested under different conditions (e.g. fed vs. starved), flies from the same vials were randomly allocated into each experimental group. In general, the experimenter was not explicitly blinded to the group or genotype.

## Acknowledgements

We thank Daisuke Hattori for MATLAB code and assistance with calcium imaging; Chris Rodgers, Barbara Noro, and Walter Fischler for advice and comments on the manuscript; members of the Axel laboratory for helpful feedback and suggestions; Adriana Nemes, Phyllis Kisloff, Miriam Gutierrez, and Clayton Eccard for general laboratory and administrative support; and Hubert Amrein, John Carlson, Ulrike Heberlein, Kristin Scott, and the Bloomington Stock Center for providing fly strains.

## Additional information

### Funding

| Funder | Grant reference number | Author |
|---|---|---|
| Howard Hughes Medical Institute | | Richard Axel |
| Simons Foundation | 54951 | Richard Axel |

The funders had no role in study design, data collection and interpretation, or the decision to submit the work for publication.

### Author contributions

Anita V Devineni, Conceptualization, Software, Formal analysis, Supervision, Investigation, Visualization, Methodology, Writing—original draft, Writing—review and editing; Bei Sun, Anna Zhukovskaya, Investigation; Richard Axel, Conceptualization, Supervision, Funding acquisition, Writing—original draft, Writing—review and editing

Author ORCIDs
Anita V Devineni  https://orcid.org/0000-0001-9540-8655
Anna Zhukovskaya  http://orcid.org/0000-0002-7096-6754
Richard Axel  https://orcid.org/0000-0002-3141-4076

Decision letter and Author response
Decision letter https://doi.org/10.7554/eLife.47677.031
Author response https://doi.org/10.7554/eLife.47677.032

## Additional files

### Supplementary files

• Supplementary file 1. Summary of statistical results. Summary of all statistical results from this study.
DOI: https://doi.org/10.7554/eLife.47677.028

• Transparent reporting form
DOI: https://doi.org/10.7554/eLife.47677.029

### Data availability

All data generated in this study are included in the manuscript and supporting files. Source data files have been provided for Figures 1-5.

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
