## [Decision Letter]

[Editors’ note: a previous version of this study was rejected after peer review, but the authors submitted for reconsideration. The first decision letter after peer review is shown below.]

Thank you for submitting your work entitled "Acetic acid activates distinct taste pathways in *Drosophila* to elicit opposing, state-dependent feeding responses" for consideration by *eLife*. Your article has been reviewed by three peer reviewers, and the evaluation has been overseen by a Reviewing Editor and a Senior Editor.

Our decision has been reached after consultation between the reviewers. Based on these discussions and the individual reviews below, we regret to inform you that your work will not be considered further for publication in *eLife*. All three reviewers found the work of high significance but felt that additional experiments would be required to support the claims.

The three main lines of essential experiments discussed by reviewers were:

1) To examine candidate receptors by testing the role of IR25a and IR76b in PER response to acetic acid (and perform calcium imaging and rescue experiments if they are required for acid responses);

2) To resolve calcium imaging discrepancies; and

3) To standardize sucrose concentrations used in behavioral studies.

We refer you to the full reviews for more detailed information.

If you feel you can fully address these concerns, we would be open to receiving a new manuscript on the topic and would make every effort to return this to the same reviewers.

*Reviewer #1:*

Devineni and colleagues report surprising dual roles of acetic acid in both appetitive and repulsive feeding behavior in *Drosophila*. Acids are generally not thought to have caloric value, yet many are part of natural food sources. Acetic acid, for example, is a by-product of many natural foods of the fly, due to fermentation of sugars by yeast and other microorganisms, yet for most animals, acids are thought to be repellants, and are generally avoided. Thus, their role has remained more enigmatic than that of most other chemicals that are sensed through the taste system.

The authors show that starved flies show concentration-dependent attraction to acetic acid (as well as propionic acid). In contrast and similar to bitter compounds, acetic acid has a repulsive effect on the attraction of sugars in well fed flies. These responses are independent of the presence or absence of olfactory organs, arguing clearly for gustatory driven responses. The attractive response is mediated by sugar sensing neurons, independent of the fatty acid response pathway that is also known to be mediated by these neurons, while the repulsive response is mediated by bitter sensing neurons. Lastly, using Ca^2+^ imaging of the neuron projections in the SEZ, the authors also show that sweet and bitter neurons are activated by acetic acid.

The paper reports a finding of general interest, namely a dual role of a fruit compound that modulates taste responses dependent on the internal state of the fly. There are, however, a number of issues that need to be addressed before the paper is suitable for publication. That includes a straightforward examination of the whether the sweet and bitter neurons responses are mediated by known sour taste receptors.

First, IRs were recently reported to mediate oviposition preference for acid containing food in females (Chen and Amrein, 2017;). First, the authors should cite this paper, as well as an earlier paper, describing a role of acids in alleviating the inhibitory effect of bitter compounds on sugars (Chen and Amrein, 2014). While oviposition is mediated by specific sour taste neurons in the tarsi, the molecular receptors (IR25a and IR76b) are broadly expressed and also found in sweet and bitter neurons of tarsi and labellum. Thus, it is straightforward to test their involvement in the reported appetitive and repulsive properties of acetic acid, which could provide a molecular receptor for the observed phenomenon and therefore would enhance the relevance of the paper.

Second, the Ca^2+^ data using acetic acids are not very consistent with PER analysis, on numerous accounts in both sugar and bitter neurons.

i) There is no dose-dependence, at least for the shown concentration of 1 and 5% in both sweet and bitter neurons (Figure 5H) even though there seems a clear increase in PER (Figure 1).

ii) Generally, the Ca^2+^ responses to acetic acid are extremely weak (compared to bitter compounds) and the authors suggest that this is due to the possibility that only a small subset of bitter neurons are involved in sensing acetic acid. Interestingly, Gr22f expressing neurons show the most robust response (Figure 5—figure supplement 3), a gene expressed in only a small subset of bitter neurons, while none of the other subsets tested showed any significant responses. Thus, the authors should test whether these neurons mediate acetic acid suppression in fed flies using neural inactivation experiments as done with Gr66a (Figure 4). But they should also consider a more complex mechanism for how acids contribute to the observed behavioral phenomenon, see next point.

Third, the model presented should consider the fact that sour taste-specific neurons not responding to sugars or bitter compounds have been identified in tarsal sensilla (Chen and Amrein, 2017). Whether or not such neurons exist in the labellum remains to be seen. Note that many labellar neurons have not yet been characterized functionally and their contribution could be tested using IR76b (or IR25a)-Gal4 tools, as the respective receptors have been implicated in sour taste (see above).

*Reviewer #2:*

Devineni et al. describe a hunger state-dependent response to acetic acid that is mediated both by bitter and sweet-sensing gustatory neurons. The authors show that acetic acid stimulates the proboscis extension response in hungry but not sated flies. This response is not pH dependent and it is not mediated by protons or acetate concentration in the solution. They continue to demonstrate that acetic acid response depends upon sweet neurons, though not the sugar receptors themselves, and inhibition (specific to the fed state) depends upon bitter neurons. Finally, the authors show acetic acid induces a calcium response at the axon terminals of sweet, and a subset of bitter neurons. The manuscript is an intriguing examination of a system that could provide insights into a state-dependent neuromodulation in the taste system. The weakest point of the manuscript is it does not clearly show that acetic acid is appetitive when flies are hungry. Authors only show PER as an indication of appetitive responses and they don't show any ingestive behavior that is induced by acetic acid. Please find my major points below.

1) In Figure 1C and 1D legend, authors say "Acetic acid was mixed with 50 mM sucrose for starved flies and with 300 mM sucrose for fed flies to induce sufficient PER to observe potential suppression." I am not sure why they chose to use 300mM sucrose in fed flies because this concentration is not tested before in the paper, or the data is not shown in Figure 1B. In addition, according to Figure 1C fed flies show 60% PER to 300mM sucrose+ 0% acetic acid and hungry flies show 100% PER to 50mM sucrose and 0% acetic acid. The starting point PER response is not the same in fed and starved flies before authors start testing acetic acid suppression on PER behavior. I think in order to compare the acetic acid suppression of PER in fed and starved flies, authors need to lower the sucrose concentration tested in starved flies to less than 10mM to equalize the PER response to sucrose in fed and starved flies before acetic acid addition. The normalization done in Figure 1D is also misleading.

2) In Figure 1E and F, authors show that flies extend their proboscis in the direction of an acetic acid stimulus. They interpret this as evidence for an appetitive response. I wonder if authors tested whether PER response to acetic acid can also trigger ingestive responses; allowing them to ingest the acetic acid solution after PER test. This will directly show that flies are extending their proboscis to acetic acid solution to feed. In addition, if acetic acid is inducing an appetitive response, then it should also enhance low concentration sugar responses which can be tested by adding acetic acid to 5mM sucrose and testing hungry flies with PER. Authors test acetic acid alone in one behavior test and make a conclusion that it is appetitive in the manuscript.

3) In Figure 3, the authors try to identify the neurons that are required for acetic acid PER. They show that silencing neurons labeled by Gr64f-Gal4 abolishes the PER mediated by sucrose and acetic acid. They also test a mutant that lack sugar sensitive GRs. These mutants show reduced responses to sucrose and enhanced responses to acetic acid. I am confused about sugar mutant results. If these mutants lack all the sugar receptors, why do they still respond to sucrose in Figure 3C? In addition, the results in Figure 3D are very confusing. Authors suggest the acetic acid responses in sugar mutants are enhanced due to either increased starvation in mutants or a mechanism that suppresses acetic acid respond in wild-type flies. Unfortunately, neither of these claims are supported by the Ca+ imaging of acetic acid response in Gr mutants shown in Figure 5—figure supplement 2.

4) In Figure 3E, authors test NorpA RNAi knockdown flies in acetic acid-induced PER. NorpA has been indicated to mediate fatty acid responses. However, recently It has been shown that fatty acids are detected by IR25a and Ir76b. (Ahn, Chen and Amrein, 2017. Is acetic acid taste mediated by the same IRs?

5) To explain the results in Figure 4, the authors mention in the text "These results demonstrate that even in the starved state, bitter neurons suppress PER to acetic acid. The observation that bitter neuron silencing has a stronger effect on acetic acid-induced PER in fed flies (Figure 4B) than in starved flies (Figure 4C) suggests that hunger inhibits the bitter pathway". I think this interpretation of the results is very weak and it is not supported by the Ca+ imaging data.

6) In Figure 5, the authors conduct Ca+ imaging experiments from sugar-sensing and bitter-sensing neurons and show that acetic acid activates both sugar sensing and bitter sensing neurons. According to single sensilla recordings published previously as authors mention in the Discussion, acetic acid inhibits sugar evoked spikes in sugar-sensing neurons and induces spikes in bitter sensing neurons. I think this contradiction between electrophysiology and Ca+ imaging is troublesome and needs to be clarified. One of the main differences between these two experiments is the electrophysiology experiments use a mixture of 100mM sucrose and varying concentrations of acetic acid and Ca+ imaging experiments in this paper are done using acetic acid alone. I think to compare these results; authors need to test Ca+ responses of sugar-sensing neurons to 100mM sucrose+ 10% acetic acid mixture in fed and starved flies.

*Reviewer #3:*

In this study of state-dependent feeding behavior in *Drosophila*, Devineni and colleagues investigate the adaptive behavioral response to acetic acid, a component of natural fly foods. Acetic acid is a source of energy at low concentrations, but toxic to *Drosophila* at high concentrations. Therefore, how valence of acetic acid is regulated by the satiety state of a fly is an exciting question of high significance in neurobiology. The authors discovered that proboscis extension response (PER) is determined by satiety state: acetic acid induces robust PER in starved flies but suppresses PER in fed flies. Imaging experiments show that acetic acid can activate both sugar- and bitter-sensing neurons in the periphery. Further genetic experiments show that acetic acid suppresses PER by activating bitter-sensing neurons and promotes PER by activating sugar-sensing neurons. The findings of this study are certainly of interest to the general readers of *eLife*. However, my enthusiasm is dampened by some concerns regarding mechanistic explanations and others of technical nature.

1) Mechanistic explanations:

The main conclusion of the study is that "Hunger shifts the behavioral response from an aversion to attraction by enhancing the appetitive sugar pathway as well as suppressing the aversive bitter pathway". First, neither the hunger signal nor the neuromodulatory mechanism has been identified. The notion that hunger increases the sensitivity of the sugar pathway and decreases the sensitivity of the aversive pathway is not supported by imaging results. Second, the current experiments cannot exclude the possibility that a third population of Gr66a- and Gr64f-negative neurons mediate the adaptive response to acetic acid. Identifying the receptor for acetic acid may help pinpoint the site of hunger modulation. Alternatively, it would be informative to know whether flies with both Gr66a and Gr64f neurons silenced exhibit any PER to acetic acid in fed or starved state.

2) Technical concerns:

There is some concern about the sensitivity of the imaging technique. In several occasions, it failed to detect behaviorally relevant tastants. For example, results presented in Figure 5D indicate that sucrose at 25 mM did not induce a statistically significant response in sugar-sensing neurons. But this concentration of sucrose is capable of inducing PER responses (Figure 5—figure supplement 5) in both fed and starved flies. In another example, Figure 5—figure supplement 2 showed that acetic acid at either 1% or 5% did not evoke a significant response in sugar-sensing neurons. But 5% acetic acid can evoke robust PER in starved flies (Figure 1A). Furthermore, based on a lack of response, it is not possible to draw the conclusion that sugar receptor mutations do not affect the acetic acid response. In addition, the response of bitter cells is problematic, as it is not significantly different from control in two of four conditions in Gr66a cells, and different only in one condition for various Gr subsets. This insensitive imaging technique may explain why the findings are inconsistent with those reported in previous publications (Inagaki et al., 2012; Marella et al., 2012; LeDue et al., 2016) that show hunger modulates gustatory responses of sugar- and bitter-sensing neurons at their axonal terminals. This reviewer notes that Inagaki et al., 2012, used the integration of ∆F/F over time to compare the responses of fed and starved flies, whereas the current study compared the peak ∆F/F. Additional analyses, concentrations, or number of animals (n=4 is low) may help resolve the inconsistencies in calcium imaging.

[Editors’ note: what now follows is the decision letter after the authors submitted for further consideration.]

Thank you for resubmitting your work entitled "Acetic acid activates distinct taste pathways in *Drosophila* to elicit opposing, state-dependent feeding responses" for further consideration at *eLife*. Your revised article has been favorably evaluated by K VijayRaghavan as the Senior Editor, Kristin Scott as the Reviewing Editor, and two reviewers.

The manuscript has been improved but there are some remaining issues that need to be addressed before acceptance, as outlined below:

1) Relevant statistics need to be included in all graphs, with stars denoting differences between different genotypes.

2) Figure 4—figure supplement 2A and B shows significant decreased responses to acetic acid in Ir25a mutants. This is consistent with a role for Ir25a in acetic acid detection. Discussion should be modified to reflect the data.

---

## [Author Response]

[Editors’ note: the author responses to the first round of peer review follow.]

Our decision has been reached after consultation between the reviewers. Based on these discussions and the individual reviews below, we regret to inform you that your work will not be considered further for publication in eLife. All three reviewers found the work of high significance but felt that additional experiments would be required to support the claims.The three main lines of essential experiments discussed by reviewers were:1) To examine candidate receptors by testing the role of IR25a and IR76b in PER response to acetic acid (and perform calcium imaging and rescue experiments if they are required for acid responses);2) To resolve calcium imaging discrepancies; and3) To standardize sucrose concentrations used in behavioral studies.

We thank the reviewing editor and the three reviewers for their thorough review of the manuscript. In our revised paper we have performed new experiments and analyses to address each of these three major points, and we believe that the manuscript is significantly improved as a result:

1) We have examined the roles of *IR25a* and *IR76b* in behavioral responses to acetic acid by testing flies carrying mutations in these genes. *IR25a* and *IR76b* mutants showed both appetitive and aversive responses to acetic acid (Figure 4—figure supplement 2), indicating that IR25a and IR76b are not required for these behaviors.

2) The reviewers raised various issues regarding calcium imaging and we address them in detail below. We agree that our initial imaging results included some puzzling observations such as a lack of dose-dependence in the acetic acid response and the fact that the neuronal and behavioral responses do not always correlate. We have performed additional experiments and analyses to address some of these issues and we have proposed possible explanations for results that remain puzzling. We have also added existing data from a second dataset where we imaged sugar and bitter neuron responses to acetic acid, effectively doubling our sample sizes. The average response values are quite similar to those in the original manuscript but the results of some statistical comparisons have changed, so we have adjusted our discussion of these experiments accordingly (see Figure 5 and the response to reviewer 3 below). Our overall conclusions from these imaging experiments are unchanged: 1) acetic acid activates both sugar- and bitter-sensing neurons, and 2) while hunger modulation of sensory neurons may contribute to the behavioral switch in the acetic acid response, the modulation of downstream neurons is also likely to play a role.

3) Reviewer 1 asked that we test fed and starved flies with acetic acid added to the same concentration of sucrose, whereas reviewer 2 asked that we test starved flies with acetic acid added to a lower concentration of sucrose. We have performed both of these experiments using multiple sucrose concentrations (Figure 1C-D and Figure 1—figure supplement 1).

Reviewer #1:[…] The paper reports a finding of general interest, namely a dual role of a fruit compound that modulates taste responses dependent on the internal state of the fly. There are, however, a number of issues that need to be addressed before the paper is suitable for publication. That includes a straightforward examination of the whether the sweet and bitter neurons responses are mediated by known sour taste receptors.First, IRs were recently reported to mediate oviposition preference for acid containing food in females (Chen and Amrein, 2017;). First, the authors should cite this paper, as well as an earlier paper, describing a role of acids in alleviating the inhibitory effect of bitter compounds on sugars (Chen and Amrein, 2014).

We apologize for neglecting to cite these very interesting papers. We felt that they were only moderately relevant because they address general effects of acids/pH, whereas the behavioral responses to acetic acid in our paper are not a general pH effect (Figure 1—figure supplement 3A); additionally, Chen and Amrein, 2014, used very low acid concentrations outside the range that we test. However, we agree that citing these studies would provide more context, and these citations are now included in the revised manuscript (Introduction, third paragraph).

While oviposition is mediated by specific sour taste neurons in the tarsi, the molecular receptors (IR25a and IR76b) are broadly expressed and also found in sweet and bitter neurons of tarsi and labellum. Thus, it is straightforward to test their involvement in the reported appetitive and repulsive properties of acetic acid, which could provide a molecular receptor for the observed phenomenon and therefore would enhance the relevance of the paper.

We thank the reviewer for this suggestion. The Amrein lab generously provided the control strain and *IR25a* and *IR76b* mutant lines used in Chen and Amrein, 2017, to show that these IRs are required for tarsal sour sensing. We found that starved *IR25a* or *IR76b* mutant flies showed robust PER to acetic acid (Figure 4—figure supplement 2A and 2D), indicating that the appetitive response in starved flies is intact. Fed *IR25a* or *IR76b* mutant flies showed low levels of PER to acetic acid that were near baseline (Figure 4—figure supplement 2B and E), indicating an intact aversive response in fed flies. Although some *IR* mutant lines showed small differences from control flies at certain concentrations (see Results and Figure 4—figure supplement 2), the fact that all lines showed appetitive and aversive responses to acetic acid indicate that IR25a and IR76b are not required for these behaviors.

Second, the Ca^2+^ data using acetic acids are not very consistent with PER analysis, on numerous accounts in both sugar and bitter neurons.i) There is no dose-dependence, at least for the shown concentration of 1 and 5% in both sweet and bitter neurons (Figure 5H) even though there seems a clear increase in PER (Figure 1).

We appreciate and agree with the reviewer’s observation that the GCaMP responses to 1% and 5% acetic acid show little to no dose-dependence. To address this issue, we imaged sugar and bitter neuron responses to a broader range of acetic acid concentrations. We have included the results of these experiments in a new figure (Figure 5—figure supplement 3) and added the following discussion to the text (Results):

“In both sugar- and bitter-sensing neurons, the average response magnitudes to 5% acetic acid were similar or slightly lower than the response to 1% (Figure 5). […] For example, secondary effects on neuronal activity could be induced by low pH or by undissociated acetic acid molecules, which may cross the cell membrane and directly affect intracellular pathways (DeSimone et al., 2001; Liman et al., 2014).”

As the reviewer mentions, it is puzzling that the dose dependence of acetic acid responses in sugar neurons does not parallel the dose dependence of acetic acid-evoked PER, which increases from 1% to 5%. This discrepancy could be explained if the bitter neurons, which inhibit acetic acid-evoked PER, showed an opposing dose dependence (with decreasing responses at increasing concentrations), but they do not. We have proposed some possible explanations in the Discussion:

“One possibility is that a subset of sugar neurons are activated dose-dependently and contributes most strongly to PER, and we may not observe this dose dependence when imaging activity across all sugar neurons. Alternatively, despite our finding that sugar neuron activity is required for PER to acetic acid, additional sensory neurons may contribute to this behavior and detect acetic acid in a dose-dependent manner.”

In addition, we note that the conditions of imaging and behavioral experiments are quite different, so flies may show different patterns of sensory activity in these two assays because they are in different states. For example, during imaging but not the PER assay, flies are head-fixed with the brain exposed, the antennae are removed, the proboscis is immobilized in an extended position, and the esophagus has been cut. Flies are allowed to consume water between trials during behavior but not imaging. We have attempted to quantify PER during calcium imaging in order to better correlate neural and behavioral responses, but this approach has not been fruitful for technical reasons.

Overall, we have now explicitly addressed the issue of dose-dependence in the Results and Discussion of the revised manuscript. We believe that the lack of canonical dose-dependence in acetic acid responses should not be considered a weakness of the paper; it is simply a feature of the data.

Moreover, it does not undermine our main conclusion from these experiments: acetic acid clearly activates both sugar and bitter neurons.

ii) Generally, the Ca^2+^ responses to acetic acid are extremely weak (compared to bitter compounds) and the authors suggest that this is due to the possibility that only a small subset of bitter neurons are involved in sensing acetic acid.

We do not entirely agree that the GCaMP responses to acetic acid are “extremely weak”. The average acetic acid responses for the experiments shown in Figure 5 were 92-166% ∆F/F_0_ for sugar neurons and 46-87% ∆F/F_0_ for bitter neurons. Moreover, acetic acid activated sugar neurons in 27 of 28 flies and bitter neurons in 31 of 36 flies, so even if the responses are considered “weak” they are present in the vast majority of flies, suggesting that they are biologically relevant. It is true that bitter neurons are more strongly activated by bitter compounds at the concentrations tested, but that does not mean that the levels of activation induced by acetic acid are irrelevant. Certainly weak levels of sensory activation could be translated into strong levels of behavior by the downstream circuit.

Interestingly, Gr22f expressing neurons show the most robust response (Figure 5—figure supplement 3), a gene expressed in only a small subset of bitter neurons, while none of the other subsets tested showed any significant responses. Thus, the authors should test whether these neurons mediate acetic acid suppression in fed flies using neural inactivation experiments as done with Gr66a (Figure 4). But they should also consider a more complex mechanism for how acids contribute to the observed behavioral phenomenon, see next point.

We thank the reviewer for this suggestion. We tested acetic acid-evoked PER in flies carrying *Gr22fGal4* and *UAS-Kir2.1* to silence *Gr22f*-expressing neurons. Their PER to acetic acid did not differ from controls (*Gr22f-Gal4/+* and *UAS-Kir2.1/+*) in either fed or two-day starved flies (data not shown). Thus bitter neurons of other classes likely contribute to the suppression of acetic acid-evoked PER even if we did not detect statistically significant GCaMP responses to acetic acid.

It is possible that other classes of bitter neurons respond weakly to acetic acid at levels that were not detectable in our experiments. In addition, in these experiments most bitter neurons showed responses to water, which could be due to minute contamination (bitter neurons are extremely sensitive), and we are only counting “significant” acetic acid responses as those that were higher than the water response. Responses that were not higher than the water response may still be biologically relevant. Moreover, we are quantifying each response as the peak ∆F/F_0_. There are certain conditions where the GCaMP traces clearly look different for water and acetic acid, but the peak response did not significantly differ (e.g. *Gr98d*-expressing cells at 1% acetic acid or *Gr22f-*expressing cells at 5% acetic acid). We have edited the figure legend and text (Results, subsection “Acetic acid activates sugar- and bitter-sensing neurons”) to clarify that our results do not rule out the possibility that these other classes of bitter neurons may also respond to acetic acid.

Third, the model presented should consider the fact that sour taste-specific neurons not responding to sugars or bitter compounds have been identified in tarsal sensilla (Chen and Amrein, 2017). Whether or not such neurons exist in the labellum remains to be seen. Note that many labellar neurons have not yet been characterized functionally and their contribution could be tested using IR76b (or IR25a)-Gal4 tools, as the respective receptors have been implicated in sour taste (see above).

We appreciate this suggestion. Having observed that IR25a and IR76b are not required for appetitive or aversive responses to acetic acid (see above), we have not further pursued the possibility that IR expressing neurons outside of the sugar- and bitter-sensing classes could be involved. However, we have now described this possibility in the Discussion (subsection “Activation of sugar and bitter neurons by acetic acid”, fourth paragraph).

Reviewer #2:[…] The weakest point of the manuscript is it does not clearly show that acetic acid is appetitive when flies are hungry. Authors only show PER as an indication of appetitive responses and they don't show any ingestive behavior that is induced by acetic acid. Please find my major points below.

We thank the reviewer for this assessment. However, the reviewer’s biggest concern that we “don’t show any ingestive behavior that is induced by acetic acid” is not accurate, since we showed that the majority of starved flies displaying PER to 5% acetic acid were willing to consume it (Results subsection “Acetic acid can elicit an appetitive or aversive taste response”, fourth paragraph; Video 2). In the example video, the fly can be seen ingesting acetic acid continuously for ~7 seconds.

1) In Figure 1C and 1D legend, authors say "Acetic acid was mixed with 50 mM sucrose for starved flies and with 300 mM sucrose for fed flies to induce sufficient PER to observe potential suppression." I am not sure why they chose to use 300mM sucrose in fed flies because this concentration is not tested before in the paper, or the data is not shown in Figure 1B.

The reason is as we stated. When we initially conducted these experiments we tried to test fed flies with acetic acid mixed into 50 mM or 100 mM sucrose, but these concentrations of sucrose evoked low levels of PER at which suppression would not be obvious. We therefore increased the concentration to 300 mM. One may note that in Figure 1B fed flies do show a reasonable level of PER to 50 mM sucrose (64%), which is why we initially thought this concentration would be sufficient. But PER can be variable across experiments, especially in fed flies, and especially given that these experiments were conducted over several years by different experimenters. (This variability is why all results shown in the same graph were always obtained from flies tested by the same experimenter on the same days.)

To address the reviewer’s concerns, we repeated these experiments and tested both fed and starved flies with acetic acid added to either 50 mM or 300 mM sucrose (Figure 1C-D). In fed flies acetic acid suppressed PER at both sucrose concentrations, but again the level of PER to 50 mM sucrose alone was so low (35%) that the PER suppression is not as clear as with 300 mM sucrose.

In addition, according to Figure 1C fed flies show 60% PER to 300mM sucrose+ 0% acetic acid and hungry flies show 100% PER to 50mM sucrose and 0% acetic acid. The starting point PER response is not the same in fed and starved flies before authors start testing acetic acid suppression on PER behavior. I think in order to compare the acetic acid suppression of PER in fed and starved flies, authors need to lower the sucrose concentration tested in starved flies to less than 10mM to equalize the PER response to sucrose in fed and starved flies before acetic acid addition.

At the reviewer’s suggestion, we tested two-day starved flies with acetic acid added to 5 mM or 10 mM sucrose (Figure 1—figure supplement 1). These flies showed a similar or lower level of PER to sucrose alone (29% at 5 mM, 55% at 10 mM) than fed flies tested with 50 or 300 mM sucrose (35% at 50 mM, 79% at 300 mM). Acetic acid still failed to suppress sucrose-evoked PER in starved flies; in fact, PER was enhanced. Thus acetic acid does not suppress PER in two-day starved flies regardless of the sucrose concentration.

The normalization done in Figure 1D is also misleading.

We have removed the graph to avoid any confusion. However, we do not agree that the original Figure 1D was misleading since the graph title clearly stated “normalized”, which means mathematically adjusting the baseline values of different groups to be equal as we had done.

2) In Figure 1E and F, authors show that flies extend their proboscis in the direction of an acetic acid stimulus. They interpret this as evidence for an appetitive response. I wonder if authors tested whether PER response to acetic acid can also trigger ingestive responses; allowing them to ingest the acetic acid solution after PER test. This will directly show that flies are extending their proboscis to acetic acid solution to feed.

Yes, we tested whether flies ingest acetic acid following PER (Results subsection “Acetic acid can elicit an appetitive or aversive taste response”, fourth paragraph 5; Video 2). The majority of two-starved flies showing PER to 5% acetic acid voluntarily ingested it, indicating that PER to acetic acid represents a component of feeding behavior.

In addition, if acetic acid is inducing an appetitive response, then it should also enhance low concentration sugar responses which can be tested by adding acetic acid to 5mM sucrose and testing hungry flies with PER.

As described above, we tested starved flies with acetic acid added to 5 mM or 10 mM sucrose (Figure 1—figure supplement 1). Acetic acid enhanced sucrose-evoked PER at both sucrose concentrations.

Authors test acetic acid alone in one behavior test and make a conclusion that it is appetitive in the manuscript.

We focus on the PER assay, but we have presented three additional lines of evidence indicating an appetitive response:

1) Flies usually extend their proboscis in the direction toward acetic acid, as is typical of appetitive but not aversive or neutral tastants (Figure 1F; Video 1).

2) Flies voluntarily ingest acetic acid following PER (Video 2).

3) Acetic acid enhances PER evoked by low sucrose concentrations (Figure 1—figure supplement 1).

3) In Figure 3, the authors try to identify the neurons that are required for acetic acid PER. They show that silencing neurons labeled by Gr64f-Gal4 abolishes the PER mediated by sucrose and acetic acid. They also test a mutant that lack sugar sensitive GRs. These mutants show reduced responses to sucrose and enhanced responses to acetic acid. I am confused about sugar mutant results. If these mutants lack all the sugar receptors, why do they still respond to sucrose in Figure 3C?

We appreciate this observation and have now mentioned it in the Results (subsection “Sugar-sensing neurons mediate PER to acetic acid”, second paragraph). The octuple sugar receptor mutants (*∆8Grs*) respond to sucrose at much lower levels than control flies, suggesting that the 8 receptors affected are the primary receptors that detect sucrose, but additional uncharacterized receptors might contribute to sucrose detection. For example, Gr43a is an internal fructose sensor that was not deleted in the *∆8Grs* line (Yavuz et al., 2015). In the labellum, *Gr43a* expression is very weak and does not overlap with sugar-sensing neurons (Miyamoto et al., 2012), which has led to the notion that it does not contribute to labellar sugar detection (Yavuz et al., 2015), but it is possible that it plays a small role when all other sugar receptors are deleted.

In addition, the results in Figure 3D are very confusing. Authors suggest the acetic acid responses in sugar mutants are enhanced due to either increased starvation in mutants or a mechanism that suppresses acetic acid respond in wild-type flies. Unfortunately, neither of these claims are supported by the Ca+ imaging of acetic acid response in Gr mutants shown in Figure 5—figure supplement 2.

There appears to be a misunderstanding here. To be clear, what we have shown is that acetic acid elicits stronger PER in the mutants than in controls but activates the sugar neurons to similar levels in fed flies. These results are not inconsistent; they simply suggest that the enhanced PER is not due to an enhanced response of sensory neurons and therefore arises downstream in the circuit. This is compatible with multiple models that we have proposed.

We cannot conclusively say that the enhanced PER in the mutants is independent of sensory neuron changes because we used fed flies for imaging, whereas PER was tested in starved flies, but these results do rule out certain models. We have edited the text (Results) to clarify our conclusions and describe the models that are most consistent with the data:

“Acetic acid elicits stronger PER in these mutants than in controls but activates the sugar neurons to similar levels in fed flies. […] Instead, the diminished response of sensory neurons to sugar as well as intensified hunger may lead to upregulation of the downstream sugar circuit and result in enhanced PER without changes in sensory neuron activation.”

To expand on the models proposed above: One possibility is that reduced activity of sugar-sensing neurons in the mutants leads to compensatory upregulation of downstream neurons in the circuit. Thus in the mutants acetic acid may activate sugar-sensing neurons at the same level as controls but downstream neurons would respond more strongly, resulting in enhanced PER. A second possibility is that themutants consume less food because they cannot taste sugar, so their hunger is intensified. Increased hunger should enhance PER to acetic acid, which could occur by upregulating either sensory or motor pathways downstream of sugar-sensing neurons.

4) In Figure 3E, authors test NorpA RNAi knockdown flies in acetic acid-induced PER. NorpA has been indicated to mediate fatty acid responses. However, recently It has been shown that fatty acids are detected by IR25a and Ir76b. (Ahn, Chen and Amrein, 2017. Is acetic acid taste mediated by the same IRs?

We have now tested the same *IR25a* and *IR76b* mutant lines that were used in Ahn et al., 2017. We found that starved *IR25a* or *IR76b* mutant flies showed robust PER to acetic acid (Figure 4—figure supplement 2A and 2D), indicating that the appetitive response in starved flies is intact. Fed *IR25a* or *IR76b* mutant flies showed low levels of PER to acetic acid that were near baseline (Figure 4—figure supplement 2B and E), indicating an intact aversive response in fed flies. Although some *IR* mutant lines showed small differences from control flies at certain concentrations (see Results and Figure 4—figure supplement 2), the fact that all lines showed appetitive and aversive responses to acetic acid indicate that IR25a and IR76b are not required for these behaviors.

5) To explain the results in Figure 4, the authors mention in the text "These results demonstrate that even in the starved state, bitter neurons suppress PER to acetic acid. The observation that bitter neuron silencing has a stronger effect on acetic acid-induced PER in fed flies (Figure 4B) than in starved flies (Figure 4C) suggests that hunger inhibits the bitter pathway". I think this interpretation of the results is very weak and it is not supported by the Ca+ imaging data.

In Figure 4 we test whether bitter neurons normally suppress acetic acid-evoked PER in the fed or starved states by silencing these neurons. We show that bitter neuron silencing dramatically increases acetic acid-evoked PER in fed flies from ~20% to ~60-80% (a 3- to 4-fold change), whereas PER in starved flies is more modestly increased from ~70-80% to ~95% (a 1.2 to 1.4-fold change). The much stronger effect in fed flies suggests that although bitter neurons suppress PER in both the fed and starved states, they exert stronger suppression in fed flies and weaker suppression in starved flies. Thus hunger must inhibit the suppressive effect of the bitter circuit on PER. Inagaki et al., 2014, arrived at the same conclusion based on the observation that bitter compounds are less effective at suppressing sucrose-evoked PER in starved flies.

There appears to be a misunderstanding regarding the interpretation of the imaging data. The reviewer seems to interpret our conclusion that “hunger inhibits the bitter pathway” as meaning that hunger suppresses the responses of bitter sensory neurons. On the contrary, we propose that hunger could suppress neuronal responses anywhere in the bitter-sensing circuit, either at or downstream of bitter sensory neurons. The fact that the responses of bitter-sensing neurons are not inhibited by hunger (Figure 5) indicates that hunger acts downstream of sensory neurons to suppress the bitter pathway. We have edited the text (Results subsection “Bitter-sensing neurons suppress PER to acetic acid”, last paragraph) to make this point more clear.

6) In Figure 5, the authors conduct Ca+ imaging experiments from sugar-sensing and bitter-sensing neurons and show that acetic acid activates both sugar sensing and bitter sensing neurons. According to single sensilla recordings published previously as authors mention in the Discussion, acetic acid inhibits sugar evoked spikes in sugar-sensing neurons and induces spikes in bitter sensing neurons. I think this contradiction between electrophysiology and Ca+ imaging is troublesome and needs to be clarified. One of the main differences between these two experiments is the electrophysiology experiments use a mixture of 100mM sucrose and varying concentrations of acetic acid and Ca+ imaging experiments in this paper are done using acetic acid alone. I think to compare these results; authors need to test Ca+ responses of sugar-sensing neurons to 100mM sucrose+ 10% acetic acid mixture in fed and starved flies.

The reviewer is correct that Charlu et al., 2013, tested mixtures of acetic acid and sucrose, but the study also tested whether sugar and bitter neurons are activated by acetic acid alone – these are the results to which we are directly comparing our data. Charlu et al. reported that acetic acid alone activated bitter neurons (Figure 2C-F), but not sugar neurons (Figure 5A), and a recent paper by Rimal et al., 2019, reported similar results. By contrast, our imaging data show that acetic acid activates the axons of both sugar and bitter neurons (Figure 5). We have added the following text to the Discussion to address this discrepancy:

“Two studies have reported that whereas acetic acid activates bitter neurons, it fails to activate sugar neurons (Charlu et al., 2013; Rimal et al., 2019). […] In addition, nonlinear amplification of spike rates into calcium responses or presynaptic facilitation by modulatory inputs could also transform weak dendritic responses into stronger axonal signals.”

Regardless of the cause of the discrepancy, we are confident in our result that sugar neurons show axonal GCaMP responses to acetic acid. Acetic acid activated sugar neurons in 27 of 28 flies initially tested (Figure 5), not including follow-up experiments which also showed consistent responses (Figure 5—figure supplement 3), and we have observed acetic acid activation of sugar neurons with three different *Gal4* drivers and two different GCaMP variants (Figure 5A-D, Figure 5—figure supplement 4, and data not shown). Moreover, the activation of sugar neurons by acetic acid is consistent with our finding that acetic acid elicits PER in starved flies and this PER is eliminated upon sugar neuron silencing (Figure 3B). If acetic acid does not activate sugar neurons, as the electrophysiology studies suggest, it would be very difficult to explain these behavioral results.

Reviewer #3:[…] My enthusiasm is dampened by some concerns regarding mechanistic explanations and others of technical nature.1) Mechanistic explanations:The main conclusion of the study is that "Hunger shifts the behavioral response from an aversion to attraction by enhancing the appetitive sugar pathway as well as suppressing the aversive bitter pathway". First, neither the hunger signal nor the neuromodulatory mechanism has been identified. The notion that hunger increases the sensitivity of the sugar pathway and decreases the sensitivity of the aversive pathway is not supported by imaging results.

There appears to be a misunderstanding here. The reviewer seems to equate hunger modulation of the sugar or bitter pathway with hunger modulation of sensory neurons. On the contrary, we propose that hunger could modulate neuronal activity anywhere in the sugar- or bitter-sensing neural circuits; hunger need not act on sensory neurons. We agree with the reviewer that the imaging data presented in the original manuscript showed that hunger does not strongly modulate sensory neuron activity, suggesting that it instead acts on downstream neurons. The data in the revised manuscript show some significant effects of hunger on sensory neuron responses, and this modulation may contribute to the behavioral switch, but downstream modulation is also likely to occur (as explained in the text). We have edited several portions of the text to make this point more clear.

Second, the current experiments cannot exclude the possibility that a third population of Gr66a- and Gr64f-negative neurons mediate the adaptive response to acetic acid. Identifying the receptor for acetic acid may help pinpoint the site of hunger modulation. Alternatively, it would be informative to know whether flies with both Gr66a and Gr64f neurons silenced exhibit any PER to acetic acid in fed or starved state.

We agree that we cannot exclude the possibility that other sensory neurons contribute to the behavioral responses to acetic acid, and we have edited the text to make this possibility more explicit (Discussion, subsection “Activation of sugar and bitter neurons by acetic acid”, fourth paragraph). We have not tested flies with both *Gr66a-* and *Gr64f-*expressing neurons silenced, and even if those flies showed residual acetic acid responses it would be unclear whether those responses are due to additional sensory neurons or incomplete silencing.

Even if other sensory neurons contribute to acetic acid responses, the observation that silencing sugar or bitter neurons largely abolishes the appetitive or aversive response to acetic acid, respectively, suggests that we have identified the primary neurons that are responsible. Moreover, the fact that behavioral sensitivity to both sucrose and lobeline differs in fed and starved flies (Figure 1B; Inagaki et al., 2012, 2014) indicates that the sugar and bitter pathways must be modulated by hunger, regardless of their contribution to acetic acid responses. Given that these pathways are also activated by acetic acid and are required for appetitive or aversive responses to acetic acid, their modulation by hunger is expected to contribute to hunger modulation of acetic acid behaviors regardless of whether other sensory neurons are involved.

2) Technical concerns:There is some concern about the sensitivity of the imaging technique. In several occasions, it failed to detect behaviorally relevant tastants. For example, results presented in Figure 5D indicate that sucrose at 25 mM did not induce a statistically significant response in sugar-sensing neurons. But this concentration of sucrose is capable of inducing PER responses (Figure 5—figure supplement 5) in both fed and starved flies.

We agree with the reviewer that in our experiments the behavioral and neural responses to tastants do not always correlate, and this may reflect a lower sensitivity of the imaging experiments as compared to the PER assay. Failing to detect responses below a certain threshold is an inherent limitation of any imaging or extracellular recording technique. However, our imaging experiments are at least as sensitive as those in other studies. For example, in Inagaki et al., 2012, the peak GCaMP responses of sugar-sensing neurons evoked by 400 mM sucrose were ~30%-50% ∆F/F_0_ in control flies (Figure 6C), whereas we observe peak responses of ~140% ∆F/F_0_ to 100 mM sucrose (Figure 5—figure supplement 3A) and ~230%-280% ∆F/F_0_ to 500 mM sucrose (Figure 5C). In bitter neurons of control fed flies, Inagaki et al., 2014, observed peak responses of ~60% ∆F/F_0_ to 0.07 mM lobeline and ~90% ∆F/F_0_ to 0.31 mM lobeline (Figure 5F1), whereas we observe peak responses of ~120% ∆F/F_0_ to 0.10 mM lobeline (Figure 5G). Thus our imaging experiments show sugar and bitter responses that are at least as high as those reported in other studies, indicating that our experiments are not less sensitive.

Aside from different sensitivities, another reason that behavioral and neural responses may not always correlate is because the conditions of these experiments are quite different. Flies may show different patterns of sensory activity in these two assays because they are in different states. For example, during imaging but not the PER assay, flies are head-fixed with the brain exposed, the antennae are removed, the proboscis is immobilized in an extended position, and the esophagus has been cut. Flies are allowed to consume water between trials during behavior but not imaging. We have attempted to quantify PER during calcium imaging in order to better correlate neural and behavioral responses, but this approach has not been fruitful for technical reasons.

In another example, Figure 5—figure supplement 2 showed that acetic acid at either 1% or 5% did not evoke a significant response in sugar-sensing neurons. But 5% acetic acid can evoke robust PER in starved flies (Figure 1A). Furthermore, based on a lack of response, it is not possible to draw the conclusion that sugar receptor mutations do not affect the acetic acid response.

There appears to be a misunderstanding here, and we have modified the figure to make it more clear. As stated in the figure legend, the asterisks indicate whether there was a statistically significant difference between control and mutant flies; they do not indicate whether the response to the tastant itself was significant. Some of the acetic acid responses were in fact significant (i.e. higher than the water response); see Supplementary file 1 for all statistics.

It is true that in this experiment the average response to each acetic acid stimulus was not always significantly different from the water response, which reflects the fact that some flies of both genotypes did not show acetic acid responses. We have edited the Results (subsection “Acetic acid activates sugar- and bitter-sensing neurons”, second paragraph) to mention this variability and analyze individual flies more closely, and we find that 6 of 9 control flies and 6 of 9 mutant flies responded to acetic acid. These data therefore support our overall conclusion that acetic acid activates sugar neurons through a mechanism that does not require sugar receptors.

In addition, the response of bitter cells is problematic, as it is not significantly different from control in two of four conditions in Gr66a cells, and different only in one condition for various Gr subsets.

We do not believe that a lack of significant responses in some conditions is “problematic”; it is simply a feature of the data. However, we have now added data to Figure 5 from a second dataset in which sugar and bitter neuron responses to acetic acid were imaged. (In the original manuscript, this dataset was mentioned in the Figure 5 legend but was not presented in the figure because the flies were tested with different sugar or bitter stimuli and the imaging conditions were slightly different. However, the peak responses to water and acetic acid in both datasets are comparable, suggesting that these datasets can be combined.) After adding these data to Figure 5D and H, the responses to both 1% and 5% acetic acid in sugar and bitter neurons are statistically significant in both fed and starved flies.

The fact that not all bitter neuron subsets respond to acetic acid also is not “problematic”, as it is entirely consistent with electrophysiology results from Charlu et al., 2013, and Rimal et al., 2019, which observed heterogeneous responses across bitter neuron classes. Charlu et al. reported acetic acid responses mainly in the S-b and I-b classes, whereas Rimal et al. reported responses only in S-a and S-b classes. Thus even studies using the same technique can obtain different results, but more importantly, our observation of significant peak responses only in the S-b class is consistent with both studies. In addition, there are conditions where the average GCaMP traces clearly look different for water and acetic acid, but the peak response did not significantly differ (e.g. *Gr98d*-expressing cells for 1% acetic acid or *Gr22f-*expressing cells for 5% acetic acid). We have edited the figure legend and text (subsection “Acetic acid activates sugar- and bitter-sensing neurons”) to clarify this point. Quantifying responses by the peak GCaMP response is a convenient way to summarize GCaMP traces, but we believe it is important to examine the traces themselves to ascertain whether there might be a real response.

*This insensitive imaging technique may explain why the findings are inconsistent with those reported in previous publications (Inagaki et al., 2012; Marella et al., 2012; LeDue et al., 2016) that show hunger modulates gustatory responses of sugar- and bitter-sensing neurons at their axonal terminals. This reviewer notes that Inagaki et al., 2012, used the integration of* ∆*F/F over time to compare the responses of fed and starved flies, whereas the current study compared the peak* ∆*F/F. Additional analyses, concentrations, or number of animals (n=4 is low) may help resolve the inconsistencies in calcium imaging.*

Again, our imaging technique is at least as sensitive as that of other studies, as we argue above. However, we have always acknowledged the possibility that there are small differences between sensory neuron responses in fed and starved flies that our experiments did not identify, especially given the high variability between flies (which other studies such as those by Inagaki et al. and LeDue et al. also observe). This caveat is stated in the Discussion (subsection “Hunger modulation of neural circuits and behavior”, second paragraph).

To address the reviewer’s concern regarding sample size, we have reanalyzed the acetic acid responses in fed and starved flies after adding data from a second dataset, as described above. We now have n = 39-43 trials, 14 flies per group for sugar neurons and n = 54 trials, 18 flies per group for bitter neurons. The average response values are similar to those in the original data, but now two comparisons are statistically significant: both sugar and bitter neurons show higher responses to 1% acetic acid in starved flies than in fed flies (Figure 5). Sugar neurons also show a trend toward higher responses to 5% acetic acid after starvation. We interpret these data as follows (Results):

“Our genetic silencing data suggest that hunger elicits a behavioral switch in the acetic acid response by upregulating the sugar-sensing circuit and downregulating the bitter-sensing circuit. Thus the enhancement of the sugar neuron response to acetic acid in starved flies may contribute to the increase in PER, but the enhancement of the bitter neuron response is not consistent with the behavioral change. This effect on bitter neurons along with the lack of significant hunger-dependent changes at 5% acetic acid in either sugar or bitter neurons, despite the fact that PER to 5% acetic acid shows even greater hunger modulation than at 1% (Figure 1A; Figure 5—figure supplement 7), suggest that sensory neuron modulation is not likely to account for the behavioral switch in the acetic acid response. The striking effects of internal state on this behavior may therefore reflect state-dependent modulation of both the sugar and bitter circuits downstream of the sensory neurons.”

Unfortunately we could not reanalyze imaging responses to sugar and bitter stimuli using the larger combined dataset because the two datasets used different concentrations or types of sugar and bitter stimuli. Separately analyzing each dataset did not reveal significant differences in the responses of fed and starved flies to sugar or bitter (n = 15-30 trials, 6-10 flies per group in dataset 1 and n = 24 trials, 8 flies per group in dataset 2).

However, in both datasets sugar neurons showed a trend toward higher sucrose responses after starvation, and this trend was quite strong for the dataset shown in Figure 5 (p = 0.063 for the effect of starvation; two-way ANOVA). When we compared the responses of fed and starved flies using the integrated ∆F/F_0_ response rather than the peak response, as the reviewer suggests, there is now a significant effect of starvation (p = 0.036 for the effect of starvation, two-way ANOVA) and a significant difference between fed and starved flies at 500 mM sucrose (p < 0.05, Bonferroni post-tests). Analyzing the integrated ∆F/F_0_ response did not reveal any significant differences between lobeline responses of bitter neurons in fed and starved flies. We have added these findings to the text (Results, subsection “Hunger modulation of taste sensory neuron responses”, first paragraph).

Despite this significant result using integrated ∆F/F_0_, we continue to use peak ∆F/F_0_ to quantify GCaMP responses throughout the paper. We believe that it is a more robust measure of neuronal activity in our experiments because, unlike the integrated response, it is not affected by how quickly the GCaMP response rises or decays during the stimulus presentation. This is especially relevant given that the exact timing of tastant delivery to the labellum can vary slightly from trial to trial. We note that Inagaki et al., 2014, and LeDue et al., 2016, also used peak ∆F/F_0_.

Finally, we note that other studies have also failed to observe strong hunger modulation of sensory neuron responses to sugar or bitter, so the lack of a significant effect would not be unprecedented. Kain and Dahunukar, 2015, did not observe a significant difference in the sugar neuron response of fed and starved flies. The data in Youn et al., 2018, also do not appear to show a clear difference between sugar neuron responses in fed and starved flies (Figure 6B, “Octopamine (-) fed, PRE” vs. “Octopamine (-) starved, PRE”), although these conditions were not directly compared. In addition, the hunger modulation effects reported in Inagaki et al., 2012, and Inagaki et al., 2014, were only present at specific concentrations of sucrose or lobeline, suggesting that they are very sensitive to the dynamic range of either the neuronal response or the GCaMP indicator, and our experiments may be operating in a different range. As we state in the Discussion (subsection “Hunger modulation of neural circuits and behavior”, second paragraph), it is possible that we would observe significant hunger modulation of sensory neuron responses if we tested a broader concentration range.

Overall, we do not believe our data are in any way less reliable than previous studies. We believe that any inconsistencies with previous studies simply reveal that hunger modulation of sensory neuron responses can vary across experiments, conditions, or concentrations, and may depend on how exactly the response is quantified.

[Editors' note: the author responses to the re-review follow.]

The manuscript has been improved but there are some remaining issues that need to be addressed before acceptance, as outlined below:1) Relevant statistics need to be included in all graphs, with stars denoting differences between different genotypes.

We have modified the figures so that all relevant statistics are depicted in the graphs, with asterisks denoting any significant differences between genotypes.

2) Figure 4—figure supplement 2A and B shows significant decreased responses to acetic acid in Ir25a mutants. This is consistent with a role for Ir25a in acetic acid detection. Discussion should be modified to reflect the data.

We appreciate this feedback, and we have added the following text to our discussion of these data (Results): “We note that both fed and starved *IR25a* mutant flies showed slightly lower PER to acetic acid than controls, suggesting that IR25a may contribute to this response even though it is not strictly required for acetic acid detection.”

We have also modified the figure legend to more explicitly point out this difference: “Two-day starved *IR25a* or *IR76b* mutant flies showed robust PER to acetic acid, although *IR25a* mutants showed a slightly lower response than *w1118* controls at certain concentrations.”